# Quantitative profiling of adaptation to cyclin E overproduction

Juanita C Limas[1], Amiee N Littlejohn[2], Amy M House[2], Katarzyna M Kedziora[3,6], Brandon L Mouery[4], Boyang Ma[2], Dalia Fleifel[2], Andrea Walens[5], Maria M Aleman[1], Daniel Dominguez[1], Jeanette Gowen Cook[1,2,4,5]

**Cyclin E/CDK2 drives cell cycle progression from G1 to S phase. Despite the toxicity of cyclin E overproduction in mammalian cells, the cyclin E gene is overexpressed in some cancers. To further understand how cells can tolerate high cyclin E, we characterized non-transformed epithelial cells subjected to chronic cyclin E overproduction. Cells overproducing cyclin E, but not cyclins D or A, briefly experienced truncated G1 phases followed by a transient period of DNA replication origin underlicensing, replication stress, and impaired proliferation. Individual cells displayed substantial intercellular heterogeneity in cell cycle dynamics and CDK activity. Each phenotype improved rapidly despite high cyclin E–associated activity. Transcriptome analysis revealed adapted cells down-regulated a cohort of G1-regulated genes. Withdrawing cyclin E from adapted cells only partially reversed underlicensing indicating that adaptation is at least partly non-genetic. This study provides evidence that mammalian cyclin E/CDK inhibits origin licensing indirectly through premature S phase onset and provides mechanistic insight into the relationship between CDKs and licensing. It serves as an example of oncogene adaptation that may recapitulate molecular changes during tumorigenesis.**

## Introduction

Cell cycle regulation depends on tight cyclin expression control to activate the cyclin-dependent protein kinase enzymes (CDK) that govern cell cycle progression (Evans et al, 1983). In G1 phase, growth factors induce cyclin D which activates CDK4 and CDK6, which in turn, inactivate the E2F inhibitor, the retinoblastoma protein (RB). E2F released from inhibition induces transcription of a suite of S phase genes, including the *cyclin E* gene. Rb hyperphosphorylation by cyclin E/CDK2 (and/or cyclin D/CDK4-6) (DeGregori et al, 1995; Dimova & Dyson, 2005; Narasimha et al, 2014; Sanidas et al, 2019) fully activates E2F (Zarkowska & Mittnacht, 1997; Yang et al, 2020).

Cells then progress from late G1 into S phase and initiate DNA replication. Importantly, cyclin E is overproduced in many cancers and tumor-derived cell lines due to gene amplification or dysregulated transcription (Barretina et al, 2012; Asghar et al, 2017; Geng et al, 2018; Chu et al, 2021), yet high cyclin E can induce replication stress, proliferation failure, and genome instability (Minella et al, 2002; Teixeira et al, 2015). The mechanisms for accommodating cyclin E overproduction are not fully understood.

Cyclin E overexpression induces premature S phase which shortens the time necessary for essential steps in the G1 phase (Ohtsubo & Roberts, 1993; Resnitzky et al, 1994; Wimmel et al, 1994; Ohtsubo et al, 1995; Spruck et al, 1999b; Matson et al, 2017). Failure to complete these steps may lead to replication stress in S phase. An essential G1 process in mammalian cells is DNA replication origin licensing, in which thousands of chromosomal sites are loaded with MCM complexes. The concerted action of the Origin Recognition Complex (ORC), and the CDC6 and CDT1 proteins renders origins competent for DNA replication (Gillespie et al, 2001; Evrin et al, 2009; Remus et al, 2009). At S phase onset, ORC, CDC6, and CDT1 are inactivated to block further MCM loading, avoiding re-licensing and re-replication. Re-replication is a form of both endogenous DNA damage and genome instability that can contribute to oncogenesis (Green & Li, 2005; Davidson et al, 2006; Arias & Walter, 2007; Zhou et al, 2020).

Because licensing is tightly restricted to G1 phase via inactivation of MCM loading factors outside G1, all licensing needed to support DNA replication in S phase must occur before the G1/S transition. One challenge in the S phase is replication fork stalling due to DNA lesions, fork damage, collisions with transcription complexes, repetitive sequences, or other barriers (Ait Saada et al, 2018; Berti et al, 2020). Stalled forks can be overcome by activating ("firing") nearby origins (Woodward et al, 2006). Cells load enough MCM during G1 to license many dormant origins, ensuring availability of both primary and dormant licensed origins (Ge et al, 2007). Without dormant origins, cells are hypersensitive to replication inhibitors, exhibit higher genome instability, and are more prone to tumor formation (Pruitt et al, 2007; Shima et al, 2007; Ibarra et al, 2008; Kawabata et al, 2011).

[1]Department of Pharmacology, University of North Carolina at Chapel Hill, Chapel Hill, NC, USA   [2]Department of Biochemistry and Biophysics, University of North Carolina at Chapel Hill, Chapel Hill, NC, USA   [3]Department of Genetics, University of North Carolina at Chapel Hill, Chapel Hill, NC, USA   [4]Curriculum in Genetics and Molecular Biology, University of North Carolina at Chapel Hill, Chapel Hill, NC, USA   [5]Lineberger Cancer Center, University of North Carolina at Chapel Hill, Chapel Hill, NC, USA   [6]Bioinformatics and Analytics Research Collaborative (BARC), University of North Carolina at Chapel Hill, Chapel Hill, NC, USA

Correspondence: jean_cook@med.unc.edu

Cyclin E overproduction in non-transformed cells perturbs normal DNA replication and can be toxic over a period of days or weeks (Minella et al, 2002; Jones et al, 2013; Kok et al, 2020), Furthermore, prolonged cyclin E overproduction can ultimately give rise to cells with altered genomes (Spruck et al, 1999a; Teixeira et al, 2015). Ectopic cyclin E overproduction in non-transformed cells shortens G1 (Resnitzky et al, 1994) and induces cells to enter the S phase with less loaded MCM than corresponding controls (Matson et al, 2017). Transient cyclin E overproduction in some cancer cells that already express high cyclin E can also inhibit licensing (Ekholm-Reed et al, 2004). It is also generally understood that CDK activity directly inhibits MCM loading factors, making origin licensing defects a likely source of cyclin E–induced genome instability (Nguyen et al, 2001; Diffley, 2004; Zielke et al, 2011, 2013). If cyclin E is toxic, how do cancer cells tolerate cyclin E overexpression and reduced origin licensing over the long timelines required for tumor development? Neither the process of adapting to the initial toxicity nor the full consequences of that adaptation have been described.

To investigate cellular adaptation to cyclin E overexpression, we analyzed multiple independent time courses as non-transformed epithelial cells adapted to ectopic cyclin E overproduction. We characterized the acute effects of cyclin E overproduction at single cell resolution and the entire process of cellular adaptation to this particular stress. We found that elevated cyclin E, but not elevated cyclin D or cyclin A, initially induces shortened G1, defects in origin licensing, and replication stress. We provide evidence that the licensing defects are likely due to truncating G1 and not direct MCM loading factor inactivation. Cells then adapt over a period of just a few weeks to lengthen G1 phase and to improve origin licensing while maintaining cyclin E expression and activity.

## Results

### Cyclin E overproduction shortens G1

To explore the cellular consequences of cyclin E overproduction we generated non-transformed human retinal pigmented epithelial cell lines with doxycycline (dox)-inducible cyclin E, cyclin D, or cyclin A cDNAs. Using non-transformed epithelial cells avoids confounding variables of genetic and epigenetic alterations present in most cancer-derived cell lines. We included cyclin D because it is overproduced in some cancers but is not associated with changes in origin licensing and replication stress. We included cyclin A because it is an alternative activator of CDK2. We established cell lines with one isoform of each cyclin (cyclin D1, cyclin E1, and cyclin A2) and isolated colonies from single cells to generate monoclonal populations. We treated cells with varying concentrations of dox for 48 h (~two cell cycles) and immunoblotted cell lysates for cyclins (Fig 1A); we quantified fold-production in Fig 1B. *CCNE1* mRNA (encoding cyclin E) in The Cancer Genome Atlas (TCGA) shows similar increases in some tumors compared with normal tissue.

Both cyclin E and cyclin A are Rb/E2F–regulated gene products (Ohtani et al, 1995; Schultze et al, 1995). Because G1-S CDK activity inactivates Rb, we tested for indirect effects of overproducing one cyclin on the others (Chellappan et al, 1991; Burkhart & Sage, 2008; Rubin et al, 2020). Immunoblotting revealed little to no consistent effect of overproducing one cyclin on expression of another cyclin (Fig 1A). We show darker exposures in Fig 1A to make endogenous cyclins visible and provide lighter exposures in Fig S6A. We selected one doxycycline dose for each cell line for further study based on the maximum dose that did not induce an immediate proliferation arrest: 100 ng/ml for cyclin D and cyclin A and 20 ng/ml for cyclin E. We tested for changes in cell cycle phase lengths using the doubling time of each cell line and the percentage of cells in G1, S, or G2/M from flow cytometry analysis (Fig S1A and B). We observed no evidence of cell death (e.g., floating cells) in response to cyclin overproduction. Overproducing cyclin E shortened G1 by nearly threefold (adjusted *P*-value < 0.0001), whereas overproducing cyclin D or cyclin A minimally shortened G1 (Fig 1C). We quantified CDK/ cyclin E–associated histone H1 kinase activity in cyclin E immunoprecipitates from lysates of asynchronously proliferating cells and quantified an average twofold increase in CDK activity (Fig 1D, N = 4 replicates quantified in Fig 1E; a lighter exposure of Fig 1D appears in Fig S6B). The fact that a large fold-increase in cyclin E protein does not induce the same fold-increase in associated

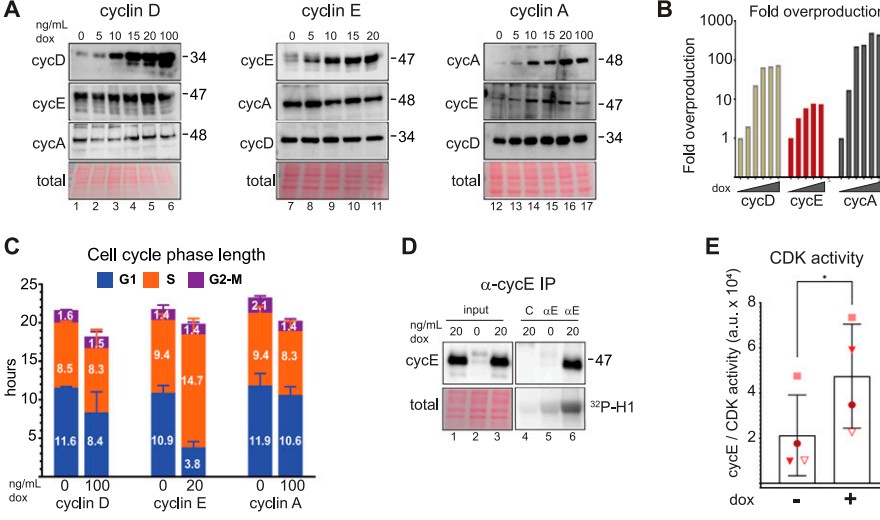

**Figure 1. Cyclin E overproduction shortens G1.**
**(A)** Non-transformed retinal pigmented epithelial (RPE1-hTert) cells with stably integrated doxycycline-inducible human cyclin D1 (left), cyclin E1 (middle), or cyclin A2 (right) were treated with doxycycline (dox) for 48 h before immunoblotting (representative of three replicates). **(B)** Fold overproduction of each induced cyclin relative to endogenous levels from (A). **(C)** Cell cycle phase lengths and distributions determined by flow cytometry and doubling time after 48 h induction with 100 ng/ml (cyclin D1), 20 ng/ml (cyclin E1), and 100 ng/ml (cyclin A2) (n = 3). G1 length difference from control: cyclin D *, cyclin E ****, cyclin A not significant. **(D)** Cyclin E and histone H1 kinase activity in immunoprecipitates of cyclin E from lysates after 48 h of cyclin E overproduction (20 ng/ml). "C" (control) is lysate from induced cells treated with non-immune serum. **(E)** Quantification of four independent replicates of (D). *P ≤ 0.05, **P ≤ 0.005, ***P ≤ 0.0005.

kinase activity may be the result having saturated other cellular factors needed for activity, such as CDK2, and/or compensatory mechanisms to restrain cyclin E/CDK2 activity.

### Cyclin E overproduction causes underlicensing and replication stress

The short G1 in cells overproducing cyclin E raised the possibility that cells entered S phase with decreased loaded MCM to license origins. To test this idea, we quantified loaded MCM per cell in early S phase using analytical flow cytometry (Fig 2A). Because MCM is unloaded throughout S phase, we focused on early S phase cells as a measure of the amount of MCM loaded in the previous G1 phase. Briefly, we pulse-labeled asynchronously proliferating cells with EdU to identify S phase cells. We then detergent-extracted to remove soluble MCM, fixed, and stained cells for loaded (or DNA-bound) endogenous MCM2 (a marker of the MCM2-7 complex) and DAPI (DNA content); see also the Materials and Methods section. Flow cytometry data are displayed as bound MCM versus DNA content with color coding as follows: MCM signal below antibody threshold (gray), cells with detectable bound MCM but no EdU incorporation (blue), and EdU-positive cells (S phase) (orange) (Fig 2B). Early S phase MCM intensity per cell indicates the abundance of licensed origins available for that S phase (green) (Figs 2C and S1A and B). Cyclin E overproduction reduced the amount of loaded (bound) MCM at S phase entry (Fig 2C, green). In contrast, cyclin D or cyclin A overproduction modestly reduced MCM loading in early S

phase (Fig S1C–F). To compare control and treated cells, we plotted the MCM intensities in early S phase cells from Fig 2C on a single histogram (loaded MCM on the x-axes). The shift of MCM loading distribution to the left in cyclin E–overproducing cells indicates S phase entry with less origin licensing (Fig 2D). Similar histograms of early S MCM loading in cells overproducing cyclin D and cyclin A showed little effect on licensing (Fig 2D). We also plotted a combination of 20 biological replicates showing single cell MCM signal intensities on one graph (Fig 2E, raw intensities in arbitrary units). We observed differences in the magnitude of underlicensing among independent replicates, but cyclin E overproduction consistently induced significant underlicensing. Across many experiments, cyclin E overproduction reduced licensing in early S by 2.5-fold.

Cells overproducing cyclin E for 2–3 d proliferated despite reduced origin licensing. We reasoned that enough primary origins were licensed in G1 to usually complete the S phase, but there would be too few licensed dormant origins to consistently suppress replication stress and genome instability. Previous work established that substantially reducing origin licensing leads to accumulation of DNA damage markers (i.e., γ-H2AX), and impaired proliferative fitness (Orr et al, 2010; Alvarez et al, 2015; Bai et al, 2016). We therefore quantified γ-H2AX as a marker of replication stress by analytical flow cytometry, using the replication inhibitor gemcitabine as a positive control. Cells overproducing cyclin E for just 48 h generated more γ-H2AX–positive cells than control cells (Fig 2F, gating strategy for scoring γ-H2AX provided in Fig S2). We also immunostained for 53BP1 nuclear bodies in Fig 4A (Chanoux

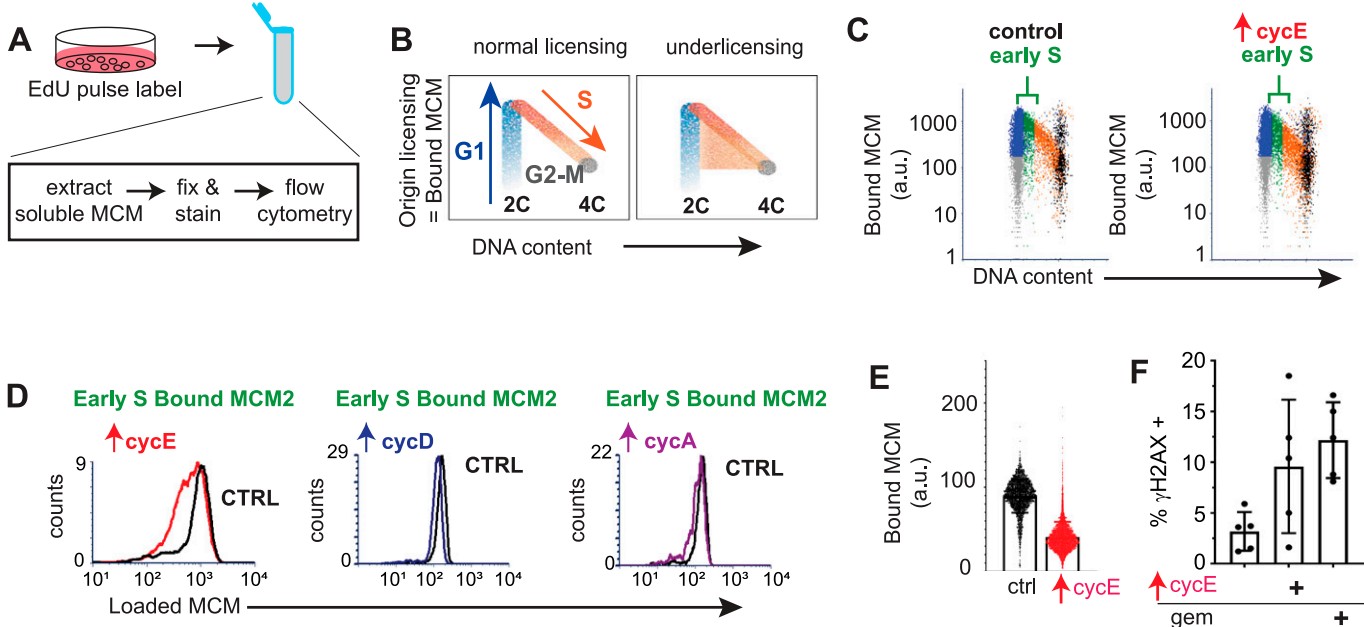

**Figure 2. Cyclin E overproduction causes underlicensing and replication stress.**
**(A)** Workflow for quantifying chromatin-bound endogenous MCM by flow cytometry. **(B)** Illustration of DNA content (x-axis), and bound MCM (y-axis). Cells are colored blue for G1 (MCM-positive), orange for S phase (EdU-positive, MCM-positive), and gray for MCM-negative. Patterns for normal licensed (left) and underlicensed (right) populations. **(C)** Analysis of MCM loading in control and cyclin E–overproducing cells after 48 h (20 ng/ml dox) detecting MCM2 as a marker of the MCM2-7 complex. Early S phase cells (defined as MCM-positive, EdU-positive, and G1 DNA content) are green and bracketed (representative of at least six replicates). **(D)** Loaded MCM in early S phase cells (intensity/cell) overproducing cyclin E (left, same cells as in C), cyclin D (middle), and cyclin A (right). Y-axis: normalized counts. **(E)** Early S phase MCM loading intensity in single cells (20 independent replicates). **(F)** Cyclin E overproduction (20 ng/ml dox) or 1 nM gemcitabine for 48 h analyzed by flow cytometry for % γ-H2AX positive cells (n = 5). Positive cells defined in Fig S2.

et al, 2009; Zeman & Cimprich, 2014). In addition, the length of the S phase increased substantially after cyclin E induction (Fig 1C and also 3D; and first reported in fibroblasts [Spruck et al, 1999a]), and we had previously documented reduced DNA synthesis rates after cyclin E overexpression in RPE1-hTert cells (Matson et al, 2017). Given the function of cyclin E in origin firing, the longer S phase likely reflects fewer origins firing from fewer licensed origins and/or checkpoint restraints triggered by replication stress. We conclude that cyclin E overproduction generates functionally "underlicensed" cells that experience replication stress.

### Proliferation and licensing perturbations during adaptation to chronic cyclin E overproduction

We wondered if cyclin E–overproducing cells could sustain long-term proliferation while severely underlicensed. We monitored cell proliferation during more than a month of continuous induction for each of the cyclin-overproducing lines by plotting days in culture (x-axis) and the inverse of the population doubling time in hours (y-axis; 1/Dt, e.g., 20 h doubling time is 0.05 on the y-axis). In the first 3 d, each of the three G1-S cyclins accelerated proliferation, but only cyclin D sustained consistent accelerated proliferation. Cells overproducing cyclin A showed only modest acceleration relative to controls (Fig S3A). In contrast, cyclin E–overproducing cells experienced an initial burst of accelerated proliferation, but it rapidly

slowed (Fig 3A). Minella and colleagues documented strong proliferation defects after 2 wk of cyclin E overproduction in human fibroblasts (Minella et al, 2002). We observed similar phenotypes in these epithelial cell lines; cultures grew more slowly after 2–3 wk of cyclin E overproduction. Surprisingly, the populations recovered after this slow proliferation period and returned to accelerated growth by weeks 4–5. Normal doubling time of these cells is 22–23 h, whereas cells overproducing cyclin E had doubling times that ranged from 15 h during the "acute" phase in the first few days to 81 h during the "proliferative crisis" after 2–3 wk. At the end, all cyclin-overproducing cultures were proliferating somewhat faster than controls though the exact timing of the proliferative crisis varied within a ~2-wk window (Figs 3A and S3A). We observed no additional proliferation rate changes after day 30 (not shown).

We were intrigued by the pattern of slowed proliferation, or "proliferative crisis," then recovery in the cyclin E–overproducing cells. We analyzed levels of induced cyclin E in adapted cells (day 39) versus cyclin E levels after just 2 d. Interestingly, adapted populations still overproduced cyclin E (Fig 3A inset, compare lanes 2 and 4), and when dox was removed for 48 h, cyclin E expression reverted to endogenous levels (Fig 3A inset, compare lanes 3 and 4). We obtained similar results for cyclins D and A (Fig S3B, compare lanes 2 and 4 and lanes 6 and 8). We analyzed several independently derived cyclin E–adapted populations, and sometimes found reduced cyclin E protein in adapted cells compared with the initial

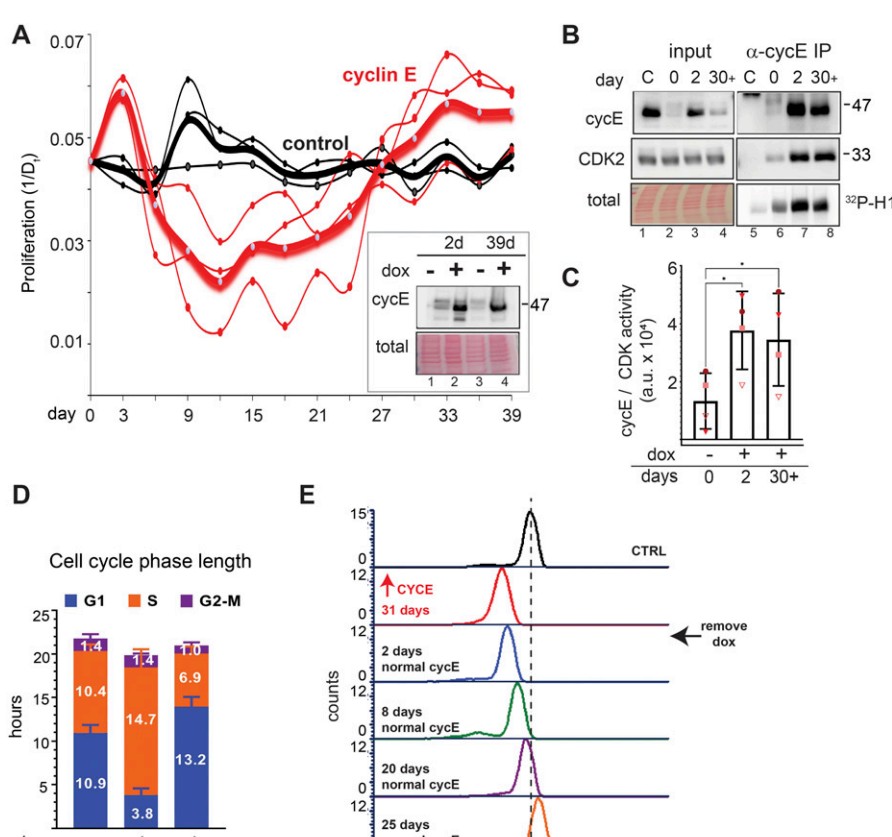

**Figure 3.  Significant proliferation and licensing perturbations from adaptation to chronic cyclin E overproduction.**
**(A)** Proliferation rate calculated every 3 d; heavy lines are means, and thin lines are three independent replicates. Proliferation plotted as the inverse of doubling time (Dt) in hours (y-axis, 1/Dt) versus days in culture (x-axis). Inset: Immunoblot for cyclin E in cells grown for 2 or 39 d in 20 ng/ml dox (lanes 2 and 4). Lane 3 is cyclin E expression in adapted cells 48 h after withdrawing dox. **(B)** Cyclin E, endogenous CDK2, and histone H1 kinase activity in cyclin E immunoprecipitates after 2 or 30+ d of proliferation (20 ng/ml dox). C is control as in Fig 1D. **(C)** Kinase activity in four replicates of (B). **(D)** Cell cycle phase lengths after 2 or 30+ d of proliferation (20 ng/ml dox, n = 3 replicates). **(E)** Early S MCM loading in control cells, adapted cells at day 31, or cells at the indicated days of culture after removing dox (representative of three replicates). Dotted line represents midpoint of control for reference.

levels after induction (e.g., Fig 3B, compare lanes 3 and 4). In all replicates, however, cyclin E levels were still higher in adapted populations than endogenous cyclin E. Importantly, the amount of CDK activity associated with cyclin E overproduction remained high in adapted cells (Fig 3B, compare lanes 7 and 8 and Fig 3C, a lighter exposure of Fig 3B appears in Fig S6C). We interpret similar kinase activity on day 2 compared with 30 or more days of induction to indicate that the amount of cyclin E overproduced in adapted cells was still enough to activate the available CDK2 even though total cyclin E had decreased. In support of adapted levels of cyclin E saturating endogenous CDK2, CDK2 protein in cyclin E immunoprecipitates was also similar (Fig 3B, compare lanes 7 and 8). We also studied cell cycle phase lengths in adapted populations compared with control and acute (2 d) populations. We again observed the extreme shortening of G1 phase as an acute early response to cyclin E overproduction, but despite their similarly elevated cyclin E–associated activity, adapted cells spent at least as much time in G1 as control cells (Fig 3D).

We hypothesized that if premature S phase entry is the primary mechanism causing underlicensing, then adapted cells with longer G1 phases would start S phase less underlicensed than cells after acute cyclin E overproduction. Alternatively if cyclin E/CDK2 directly inactivates MCM loading then the high cyclin E/CDK2 activity in adapted cells would still induce strong underlicensing. We thus analyzed MCM loading in adapted cells, and surprisingly these cells had intermediate licensing. Adapted cells were not as underlicensed as cells with very short G1 phases (Fig 2), but still underlicensed relative to uninduced cells (Fig 3E, compare control with ↑CycE 31 d). As others have shown, intermediate licensing can be compatible with robust cell proliferation (Woodward et al, 2006; Ge et al, 2007; Ibarra et al, 2008).

To test if the intermediate licensing was solely dependent on elevated cyclin E, we withdrew dox and monitored licensing in adapted cells (Fig 3A inset, compare lanes 3 and 4). Cells proliferated normally with no second proliferative crisis, indicating that adaptation did not include acquiring dependence on high cyclin E (not shown). Surprisingly, returning cyclin E to endogenous levels after 30 d of adaptation to high cyclin E did not immediately revert to full licensing, despite the rapid return of cyclin E to normal levels (Fig 3A). We analyzed cells at 2, 8, 20, and 25 d after dox withdrawal and quantified a gradual improvement in origin licensing over the course of this recovery period (Fig 3E, compare adapted cells after 31 d of dox induction [red] to days 20 and 25 after dox removal [purple and orange]). This finding suggests that adaptation is associated with physiological changes independent of constant cyclin E overproduction. Furthermore, the fact that cells eventually returned to normal licensing indicated that adaptation was not solely through selection for permanent genetic changes, although mutations may have occurred. We also did not observe rare clones arising from a field of mostly arrested cells during the proliferative crisis.

### Chronic cyclin E overproduction induces transient DNA damage response markers and heterogeneous cell cycle arrest

We next sought to analyze replication stress markers. We anticipated that cells initially accumulate endogenous DNA damage due to chronic underlicensing and replication stress. To test this notion, stained for the replication stress and DNA damage marker, 53BP1, every 3 d during chronic cyclin E overproduction (Schultz et al, 2000). We analyzed a single population of cells for 33 d by quantifying the number of 53BP1 nuclear bodies in thousands of cells from 20 fields of view in each time point (Fig S4A). We chose a conservative arbitrary threshold for automated counting to quantify the brightest 53BP1 nuclear bodies (see the Materials and Methods section). We classified cells as having one nuclear body, multiple nuclear bodies, or no nuclear bodies. Before induction, cells produce some 53BP1 nuclear bodies due to constitutive replication stress in unperturbed cells (Moreno et al, 2016). Within the first few days of cyclin E overproduction, we consistently found cells with one or more 53BP1 nuclear bodies which eventually subsided to control levels (Fig 4A). The period of elevated 53BP1 nuclear bodies coincided with slowed proliferation, and the return to baseline 53BP1 signals coincided with the return to accelerated proliferation in Fig 3A.

The increase in 53BP1 nuclear bodies indicated DNA damage that could trigger a cell cycle checkpoint contributes to the proliferative crisis. We therefore collected lysates every 3 d during the adaptation to probe for the induction of p53 and its downstream target, p21, both well-established markers of the DNA damage response (El-Deiry et al, 1993; Macleod et al, 1995; Kubbutat et al, 1997). We detected a minor and sustained elevation in p53 protein levels throughout most of the time course, but a marked induction of p21 (Fig 4B). The induction of the p21 CDK inhibitor preceded the proliferative crisis and lasted past the return to rapid proliferation in this example. In contrast, the p27 CDK inhibitor was unaffected. The p21 protein is degraded in S phase cells, but p21 is stable in G1 cells (Abbas et al, 2008; Nishitani et al, 2008). Because adapted cells spend more time in G1 and less time in S phase (Fig 3D), elevated p21 levels in lysates at later time points could reflect cell cycle phase distribution, a persistent DNA damage response, or both. On the other hand, p21 induction at earlier time points is more likely to be attributable to DNA damage because G1 phases were shorter and 53BP1 nuclear bodies more abundant at those times. This early p21 induction may have limited the amount of kinase activity the overproduced cyclin E could stimulate (Figs 1E and 3C). We also detected slightly stronger p21 induction from the addition of neocarzinostatin (NCS), a DNA damage–inducing agent, in cells after 2 d of cyclin E overproduction compared with adapted cells (Fig S4B). Our working model is that adaptation includes a long, but transient activation of the DNA damage response.

Independent adaptations varied in the dynamics of the proliferation responses (Fig 3A). Nevertheless, the pattern was consistent in each replicate: cultures initially proliferated rapidly, then very slowly, and then returned to rapid proliferation. We routinely monitored cultures in proliferative crisis and observed heterogeneous morphologies: some large cells reminiscent of senescence, some elongated cells, and others resembling normal proliferating cells (not shown). We were therefore interested in understanding adaptation in single cells. The biochemical kinase activity analysis in Figs 1 and 3 could not distinguish cyclin E–associated activity in G1 cells from activity in other cell cycle phases. To further examine single cell CDK activity in real time we introduced a single cell reporter for both CDK1 and CDK2 activity via nuclear-to-cytoplasmic

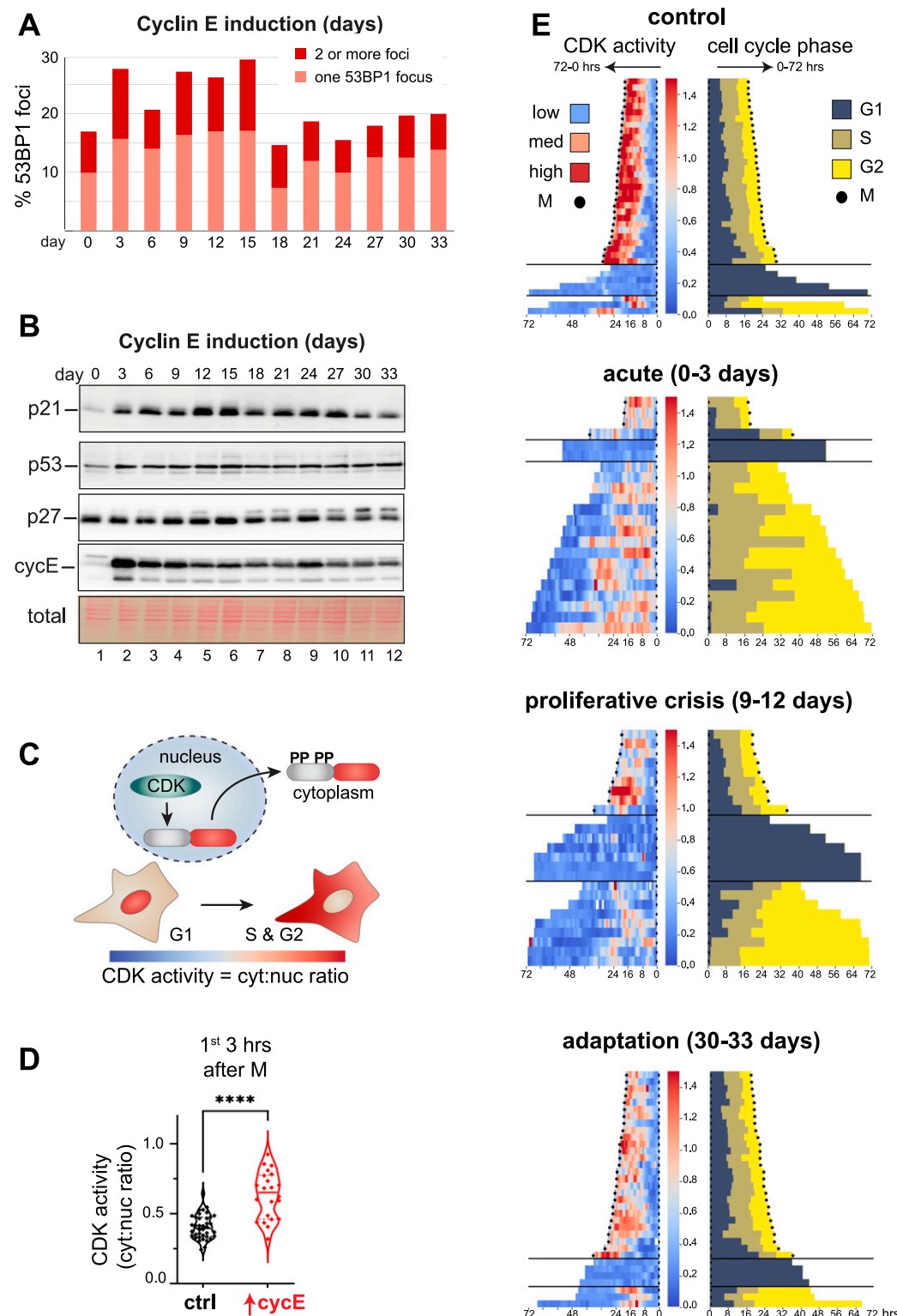

**Figure 4. Chronic cyclin E overproduction induces transient DNA damage response markers and heterogeneous cell cycle arrest.**
**(A)** Cells overproducing cyclin E (20 ng/ml dox) were stained for endogenous 53BP1 nuclear bodies at the indicated time points; cells were classified as having one or more than one bright 53BP1 nuclear body. **(B)** Endogenous p21, p27, and p53 plus both ectopic and endogenous cyclin E from cells treated as in (A). **(C)** Illustration of the translocation-based CDK1/CDK2 activity biosensor (Spencer et al, 2013). **(D)** CDK activity in the first 3 h of G1 after cyclin E induction plotted as the cytoplasmic to nuclear ratio of biosensor localization (n = 60 cells ****P ≤ 0.00005). **(E)** CDK activity (left) and cell cycle phase lengths defined by PCNA localization (right) during 72 h live imaging at the indicated times after continuous cyclin E overproduction. Individual cell traces begin with cell division; left is CDK activity and right is cell cycle phases. Low and high CDK activity via biosensor localization are blue and red respectively. Cell cycle phase transitions are defined by changes in PCNA localization. Black dots indicate mitosis.

translocation upon CDK-mediated phosphorylation (Fig 4C). This reporter was generated from a fragment of the DHB CDK substrate, does not respond to CDK4/6, and has been extensively characterized (Gu et al, 2004; Hahn et al, 2009; Spencer et al, 2013; Schwarz et al, 2018; Yang et al, 2020). In addition, we introduced a PCNA-based reporter to mark the boundaries of S phase by changes in localization (Leonhardt et al, 2000). To monitor CDK1/2 activity, we measured the cytoplasmic-to-nuclear fluorescence intensity ratio of the reporter: a low ratio indicates low CDK activity, a high ratio indicates high CDK activity. We first tested the immediate effects of cyclin E induction by analyzing G1 cells in the first 3 h after mitosis shortly after doxycycline addition. As expected, high cyclin E induced a strong spike in CDK1/2 activity evidenced by increased cytoplasm-to-nucleus fluorescence (Fig 4D).

We then subjected this reporter line to chronic cyclin E overproduction and collected time lapse fluorescence images for 72 h at four distinct time points: (1) control, (2) "acute" cyclin E overproduction (0–3 d), (3) proliferative crisis (9–12 d), and (4) adaptation 30–33 d. For individual cells, we calculated CDK activity and cell cycle phase lengths or arrest, restricting analysis to cells with a visible mitosis and excluding cells that did not divide at all during the entire 72 h. Fig 4E is heat maps of CDK1/2 activity (left) and tracks for cell cycle phases (right); each trace begins with mitosis and either ends with mitosis or the end of imaging (Fig 4E). In the control population, CDK1/2 activity followed the expected pattern: low in G1 (blue), increasing through S and G2 phases (pink and red), followed by mitosis (black dots) for most cells.

Although these cultures started from monoclonal populations, we found wide intercellular heterogeneity in CDK1/2 and cell cycle dynamics. At each time point we identified three discrete patterns: (1) a complete cell cycle that ended with mitosis, (2) mitosis followed by a G1/G0 arrest, and (3) cells that appeared to complete S phase but never divided. During first 3 d of cyclin E overproduction ("acute") many cells had extremely short G1 phases (dark blue bars) and long S phases, as expected from flow cytometry analysis in Fig 3D, but also less pronounced oscillations in CDK activity compared with controls. Many cells completed the S phase but never entered mitosis or underwent nuclear envelope breakdown. Instead, these "G2-arrested" cells returned to low CDK activity as though they had skipped mitosis, but they did not start S phase again during the imaging window (Fig 4E, acute [0–3 d]).

In contrast, many individual cells in proliferative crisis divided but then entered very long G1 phases and neither increased CDK activity nor progressed to the S phase. Substantial numbers completed the S phase but arrested in the subsequent G2 with low CDK activity (Fig 4E, proliferative crisis [9–12 d]). Some cells were non-dividing with no visible mitoses for the entire 72 h (not shown). Nonetheless, a fraction of cells completed a full cell cycle despite altered patterns of CDK activity (Fig 4E, top rows of "acute" and "proliferative crisis"). After 30 d of chronic cyclin E overproduction, cell cycle phases were more typical of control cells, although interestingly, CDK activity was lower in S and G2 phases compared with controls (Fig 4E, adaptation, 30–33 d).

In addition, we examined all the cells in the final frame of each movie by automated image analysis at the end of each 3 d of imaging and plotted the CDK1/2 activity for each cell (Fig S4C). Based on prior observations (Spencer et al, 2013; Schwarz et al, 2018)

cells with a cytoplasmic:nuclear reporter ratio of 0.75 or higher typically are committed to S phase entry or are already in the S or G2 phase. Although activity was high very early (first 3 h) after cyclin E induction (Fig 4D), by the end of 3 d of cyclin E overproduction, the population with high CDK1/2 activity had already substantially dropped and remained low during the proliferative crisis (Fig S4C, ~days 9–12). We expect that if adapted cells were primarily the descendants of rare mutants, we would have observed mostly arrested or dead cells and rare clones expanding during and after the proliferative crisis. Instead, we observed dispersed proliferating cells of different proportions throughout the culture at all time points.

Taken together, we conclude that chronic cyclin E overproduction induces replication stress and DNA damage that accumulates over many cell divisions, ultimately inducing a transient proliferative crisis. During this crisis, individual cells may permanently arrest in either G1 or G2. G2-arrested cells can skip mitosis and persist in a G1/G0–like state with low CDK activity. A subset of cells continued to proliferate with altered CDK dynamics, establishing a new population that adapted to cyclin E overproduction.

### Transcriptome shifts in cell cycle–regulated genes during adaptation to cyclin E overproduction

Cells that adapted to cyclin E overproduction proliferated slightly faster than controls, yet had longer G1 phases and intermediate origin licensing compared to cells during the initial response to cyclin E. In the absence of cyclin E overproduction, cells maintained intermediate origin licensing through many cell divisions (Fig 3E). We hypothesized that adaptation might involve gene expression changes, so we performed RNA sequencing to identify such changes in three independent replicates from five time points: day 0 ("control" without induction), day 2 ("acute"), day ~9–15 ("proliferative crisis"), or day ~35 ("adapted") after continuous cyclin E induction. Because adaptation dynamics varied among replicates, we collected RNA from cells when proliferation was slowest and when it had fully rebounded (see the Materials and Methods section). In addition, we withdrew doxycycline and allowed cells to proliferate for another 20 d ("recovery").

We first examined *CCNE1* mRNA across time points and within replicate groups. Although samples were collected independently from adaptations that were initiated weeks apart, the levels of *CCNE1* mRNA were very similar among replicates (Fig 5A). After 2 d of acute induction, *CCNE1* mRNA abundance increased more than 20-fold, similar to the 10-fold induction of cyclin E protein in Fig 1A (compare lanes 7 and 11). *CCNE1* mRNA levels decreased during proliferative crisis and in adapted cells to an intermediate level that was still significantly higher than endogenous *CCNE1* (Fig 5A, C, and D). The mechanism of down-regulation was not loss of the tet repressor mRNA (not shown), but could be epigenetic partial silencing of the transgene, or changes in mRNA processing or stability. We presume this mRNA down-regulation after the acute phase is driving partial cyclin E protein down-regulation in Fig 3A (inset, lane 4) and Fig 4B. As expected, *CCNE1* mRNA returned to endogenous levels after doxycycline withdrawal (Fig 5A).

Surprisingly, we detected relatively few significantly differentially expressed genes across time points (using a –log(false discovery

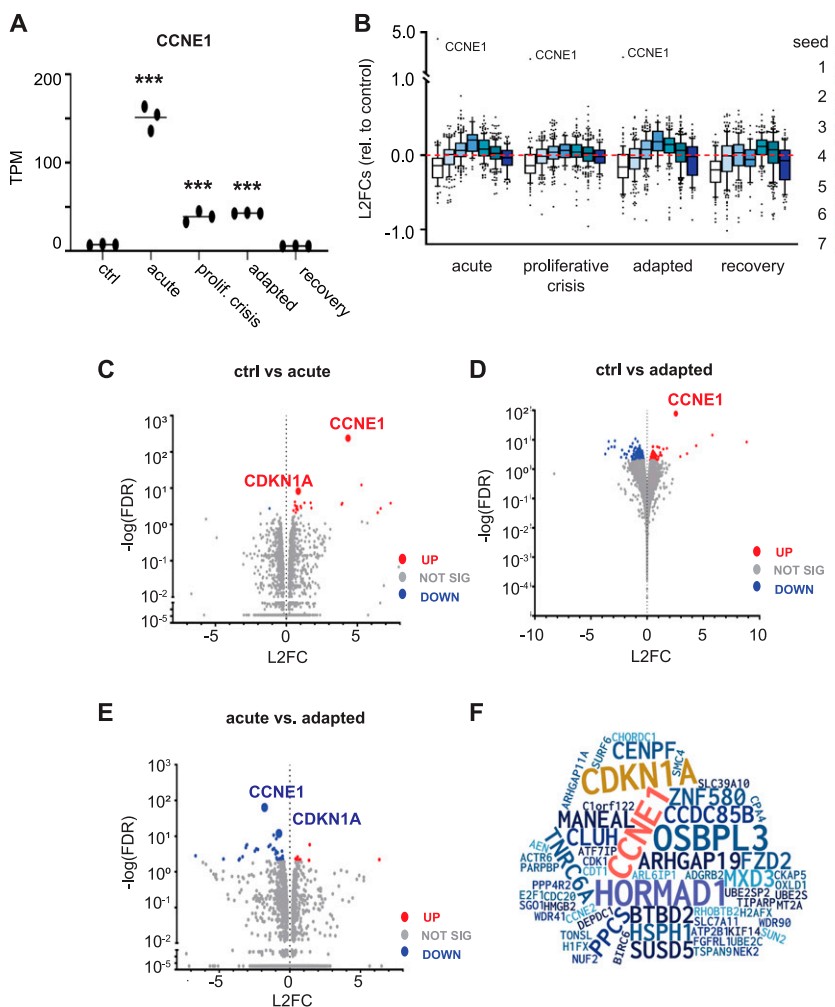

Figure 5.  Transcriptome shifts in cell cycle-regulated genes during adaptation to cyclin E overproduction. (A) Cells were harvested for mRNA analysis at the indicated times during cyclin E overproduction (20 ng/ml) and 2 wk after doxycycline removal. Ctrl = no dox, acute = 2 d induction, proliferative crisis = 9–15 d induction at the point of lowest proliferation, adapted ≥ 30 d induction, and recovery = 2 wk after doxycycline was removed from adapted cells. CCNE1 expression in transcripts per million (TPM) at each time point (n = 3 independent biological replicates). * indicates adjusted P-values of differential expressions compared with day 0 for CCNE1 based on DESEQ. *$P \leq 0.05$, **$P \leq 0.005$, ***$P \leq 0.0005$. (B) Log$_2$ fold change (L2FC) relative to uninduced control (day 0) of CCNE1 and the genes in seven periodic seed curves defined in Dominguez et al (2016). Seed 1 genes peak in G1 and seed 6 and 7 genes peak in M phase-M/G1 transition; box-plots mark the medians and interquartile ranges. (C) Volcano plot of all genes comparing control to day 2 after induction (acute); statistically significant genes determined by L2FC > 2 are red (up-regulated) and blue (down-regulated). Statistically insignificant genes (L2FC < 2) are gray. (D) As in (C) comparing control to adapted cells. (E) As in (C) and (D) comparing acute to adapted cells. (F) Word cloud of all statistically significant genes in any pairwise comparison; letter sizes correspond to the frequency of genes in all possible comparisons. See also Table S1.

rate) ≥ 2; 426 genes). Given the robust impact of CCNE1 induction on cell cycle-associated phenotypes (Figs 1–3), we focused our analysis on previously defined cohorts of genes whose expression oscillates with the cell cycle (Dominguez et al, 2016). We assessed differentially expressed genes relative to controls for groups of genes assigned to seven periodic seed curves that peak progressively during the cell cycle. Seed 1 genes have peak expression in G1, seed 4 genes peak in G2 phase, and seeds 6 and 7 genes peak in M phase and the M/G1 transition, respectively. We plotted the log$_2$ fold changes (induced versus non-induced) for each of the seven gene cohorts at each time point. We noted that the G1 cohort (seed 1, white) was clearly down-regulated in cyclin E–overproducing cells with a corresponding increase in seed 4 (G2) genes (Fig 5B). Expression of seed 1 genes remained low even after dox withdrawal. These shifts in expression, particularly for G1-expressed genes, may reflect cell cycle phase length changes, may be the driving cause of those changes, or both.

The CDKN1A gene encoding the CDK inhibitor p21 was among the few genes whose expression significantly changed comparing control to day 2 (acute overproduction, Fig 5C), and comparing day 2 to day 35 (acute versus adapted cells, Fig 5E). Elevated CDKN1A

mRNA is consistent with the elevated 53BP1 nuclear bodies and p21 protein levels during both the acute response and proliferative crisis when markers of replication stress were highest (Fig 4A and B).

To investigate potential operant pathways in cells adapting to cyclin E overproduction, we performed Gene Set Enrichment Analysis with our transcriptome data. Acute cyclin E overproduction caused down-regulation of several signaling pathways with the pathways for early and late estrogen response predominant as down-regulated pathways during proliferative crisis and adaptation (Fig S5A). We found few statistically significant up-regulated pathways (Fig S5B), and few statistically significant individual genes were up-regulated between control and proliferative crisis cells (Fig S5C). However, when we compared all genes at the beginning (control) and end of the adaptation, we found that selected genes involved in origin licensing (MCM5 and CDT1) were down-regulated and DNA damage genes were either up-regulated (ATM) or down-regulated (H2AFX), although these individual gene changes fell short of our criteria for statistical significance (Fig S5D). Last, to better visualize the impact on cell cycle genes over the course of adaptation we constructed a word cloud of statistically significant

changed genes in any pairwise comparison (Fig 5F); all significant gene expression changes relative to controls are provided in Table S1.

## Parallel replication stress in breast cancers with high *CCNE1* expression

We next assessed *CCNE1* expression in human breast tumors (TCGA). As expected, *CCNE1* mRNA correlates with *CCNE1* copy number changes, and tumors harboring *CCNE1* amplifications expressed ~20-fold higher *CCNE1* mRNA than diploid tumors (Fig 6A, see the Materials and Methods section). This level of *CCNE1* overexpression in primary tumor samples is similar to the fold-induction of *CCNE1* in Fig 1. Moreover, associated DNA damage response markers (i.e., phospho-CHK1), also correlated with *CCNE1* expression in the TCGA database (Fig 6B) and in another analysis of human biopsies (Guerrero Llobet et al, 2020), consistent with elevated cyclin E–induced replication stress. We are thus confident that we recapitulated some of the consequences of cyclin E overproduction that are visible in human tumors. This experimental approach can shed light on both the early stress induced by oncogene expression and the changes that arise to accommodate that stress.

# Discussion

Cyclin genes are frequently overexpressed in human cancers either from copy number changes or loss of normal upstream signaling control (Donnellan & Chetty, 1999; Chu et al, 2021). The effects of cyclin overproduction on proliferation and genome stability are difficult to dissect in fully developed cancer cells because the cyclin changes are combined with many genetic and epigenetic changes that activate growth signaling, inhibit apoptosis, and impair checkpoints (Hanahan & Weinberg, 2011). We analyzed cells throughout adaptation for changes in cell cycle phase length, markers of replication stress, and CDK activity using a variety of single-cell methods. This approach differs from comparing starting cell populations with resistant or fully adapted populations.

To isolate individual effects of G1–S cyclin overproduction from other changes, we produced each cyclin under inducible control in non-transformed epithelial cells. Cyclin D1 overproduction induced slightly faster proliferation and shorter G1 phases but did not cause a proliferative crisis as cyclin E overproduction did. Cyclin E induced a much shorter G1 phase than either cyclin D or cyclin A. There may be a minimum G1 length below which cells cannot proliferate well. Cyclin E overproduction triggered premature S phase entry, in part, because unlike cyclin D, direct cyclin E/CDK2 substrates include those required for origin firing (Boos et al, 2011; Kumagai et al, 2011; Siddiqui et al, 2013). Cyclin D overproduction may simply mimic growth factor signaling but only indirectly induce S phase entry. Cyclin A does not appear to be rate-limiting for S phase entry in these cells/under these conditions, perhaps because of degradation by APC/C in G1 (Geley et al, 2001).

We propose that a major contributor to chronic replication stress in cyclin E–overproducing cells is insufficient origin licensing. We acknowledge that other mechanisms may also contribute to

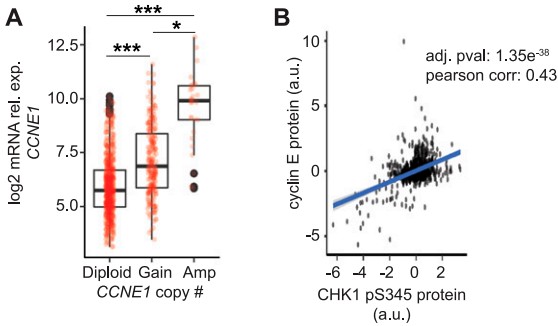

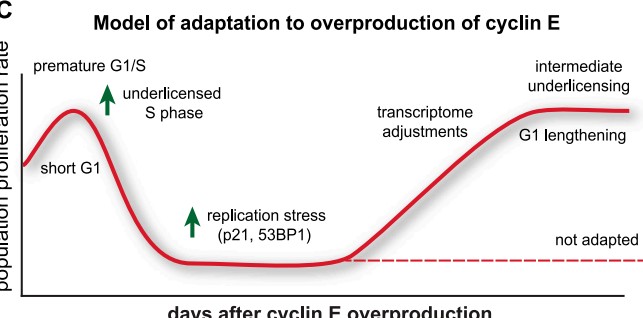

**Figure 6.  Cyclin E overproduction in RPE1-hTERT cells is representative of cyclin E–overexpressing breast cancers.**
**(A)** *CCNE1* expression relative to gene copy change in The Cancer Genome Atlas (TCGA) breast tumors. Number of cases in TCGA for *CCNE1*: Diploid (601), Gain (232); Amplification (29) *$P \leq 0.05$, ***$P \leq 0.0005$. **(B)** Phospho-CHK1 relative to cyclin E protein in human breast tumors from the TCGA database. **(C)** Model for adaptation to cyclin E overproduction: Initially a short G1 leaves insufficient time for origin licensing. These cells experience replication stress, DNA damage, and slow or arrested proliferation. With continued cyclin E overproduction, a subset of cells eventually adapt, lengthen their G1, and proliferate with perturbed cell cycle gene expression.

cyclin E–induced replication stress. For example, premature S phase can also induce replication-transcription collisions by allowing intragenic origins to fire that normally would not (Macheret & Halazonetis, 2018). Previous studies have also documented the genotoxic effects of cyclin E overproduction, but did not explore origin licensing as a contributor (Spruck et al, 1999a; Minella et al, 2002). Others have also observed less MCM loading in cyclin E–overproducing cells but did not examine long-term proliferation effects and suggested direct CDK-mediated MCM loading inhibition as a mechanism (Ekholm-Reed et al, 2004; Matson et al, 2017). The direct inhibition paradigm was first established in budding and fission yeast studies where MCM loading proteins are inhibited by CDK-mediated phosphorylation (Nguyen et al, 2001; Diffley, 2004). In mammalian cells MCM loading proteins are also inhibited during S phase to prevent inappropriate re-licensing, but these inhibitory mechanisms may be attributable to cyclin A more than cyclin E. For example, CDT1 is degraded by the SCF[SKP2] E3 ubiquitin ligase after phosphorylation, but CDT1 only binds cyclin A and not cyclin E (Zhou et al, 2020). Cyclin E is also not required for CDT1 degradation during replication in *Xenopus laevis* extracts (Arias & Walter, 2005). Similarly, the largest subunit of ORC, ORC1, is degraded in the S phase but is a much better substrate for cyclin A/CDK2 than it is for cyclin E/CDK2 (Méndez et al, 2002). Human CDC6 is translocated to the cytoplasm during S phase after

phosphorylation mediated by cyclin A, but not by phosphorylation at the cyclin E–dependent site at serine 54 (Petersen et al, 1999; Yim et al, 2013). Importantly, cyclin E *activates* rather than inhibits mammalian CDC6 because phosphorylation at serine 54 stabilizes CDC6 during late G1 phase (Mailand & Diffley, 2005). We assert that unlike yeast CDKs that directly *inhibit* MCM loading proteins, mammalian cyclin E is not a strong *direct* inhibitor of origin licensing, and that MCM loading factor inactivation is largely dependent on other CDKs.

Cyclin E/CDK2 activity rises in late G1 phase at the time that origin licensing is most active (Mailand & Diffley, 2005). It would be counterproductive for cyclin E to block origin licensing in G1. Cyclin A, however, is only expressed once S phase begins. The strong underlicensing we demonstrate in cyclin E–overproducing cells is more likely attributable to the *indirect* effects of premature S phase onset. Activating DNA replication triggers CDT1 degradation once PCNA is loaded at replication forks and recruits the CRL4$^{CDT2}$ E3 ubiquitin ligase (Arias & Walter, 2006; Hu & Xiong, 2006; Senga et al, 2006). Because MCM loading requires CDT1 (Maiorano et al, 2000), early S phase onset from cyclin E overproduction stops further MCM loading. This degree of underlicensing causes accumulating replication stress that induces the proliferative crisis. Of note, cyclin A overproduction had minimal effects on G1 length (Fig 1C) and induced modest underlicensing (Figs 2D and S1F). This limited inhibition could reflect direct MCM loading factor inhibition in G1 by cyclin A–mediated phosphorylation. It is possible that at high levels, cyclin E may inactivate MCM loading proteins directly, contributing to the moderate underlicensing in adapted cells. One prior study identified a CDK2 phosphorylation site in MCM7 phosphorylated by both cyclin E and A in vitro that may interfere with MCM2-7 complex formation and chromatin loading (Wei et al, 2013). Together, our data furthers defines the relationship between CDK activity and licensing in mammals and the importance of G1 length to avoid replication stress.

Remarkably, adaptation to high cyclin E was not just a reversal of the overproduction itself. Adapted cells still overproduced cyclin E, and CDK kinase activity stayed elevated relative to the starting population. If a major cause of the replication stress was underlicensing due to truncating G1, then the longer G1 phases that developed in adapted cells likely provided enough time for adequate origin licensing. Indeed, adapted cells were substantially less underlicensed than cells experiencing acute cyclin E overproduction. These observations are consistent with high cyclin E–expressing cancer cell lines proliferating well (Ekholm-Reed et al, 2004; Asghar et al, 2017; Geng et al, 2018).

The mechanism(s) by which G1 lengthened may include down-regulating the cohort of G1-induced genes, possibly by deactivating one or more components of the upstream signaling and regulatory systems that induce these genes. Notably, the estrogen response pathway was significantly down-regulated in adapted cells. Future proteomic studies will shed additional light on these adaptation mechanism(s). Interestingly, not all cells with short G1 phases are underlicensed. A hallmark of pluripotency is a very short G1, and we previously demonstrated that pluripotent stem cells license origins rapidly to achieve high MCM loading (Matson et al, 2017).

Overall increased p21 protein levels in fully adapted cells (Fig 4B) may also have promoted G1 lengthening. Despite cyclin E–associated kinase activity remaining elevated (measured via immunoprecipitation from asynchronous populations) (Fig 3C), the dynamics of CDK activation in individual cells was clearly perturbed and appeared more delayed (Fig 4E). Induction of DNA damage and replication stress markers (53BP1 and p21) likely contributed to the proliferative crisis, and their down-regulation contributed to eventual adaptation. It is not surprising that p21 levels were induced in cells experiencing high replication stress and proliferative crisis because of its involvement in DNA damage responses (El-Deiry et al, 1993; Macleod et al, 1995).

Adaptation is typically associated with selection for genetic changes. Previous studies of transient cyclin E overproduction documented genomic deletions in individual cells (Teixeira et al, 2015). Cells chronically overproducing cyclin E did not appear to adapt by the expansion of rare mutant clones, but rather by nongenetic perturbations. We cite three observations in support of this interpretation. First, we routinely inspected cultures during adaptation and found proliferating cells at all time points without widespread arrest. Second, cultures consistently overcame proliferative crisis within two to 3 wk regardless of the starting population size. Third, adapted cells had intermediate origin underlicensing and gradually returned to normal licensing over a period of weeks without cyclin E overproduction. A mutation would not revert in the absence of any evidence for selective pressure. We acknowledge that contributions from mutations cannot be completely ruled out, particularly as cells return to proliferation from crisis. We speculate that the primary adaptation mechanism is epigenetic, signaling rewiring, post-transcriptional/translational adjustments or some mosaic combination of these or other nongenetic changes.

In conclusion, our unique study characterizes the process of adaptation to cyclin E overproduction at single-cell resolution. Adaptation was consistently rapid, resulting in cells with altered growth and replication phenotypes. Our model for adaptation to cyclin E overproduction is that high cyclin E/CDK2 activity causes a short G1, underlicensing, and accumulating replication stress coupled with DNA damage (Fig 6C). This stress triggers slowed proliferation with cell cycle patterns and arrests that vary among individual cells. After several generations, the proliferating population has longer G1 phases, shifts in gene expression, intermediate licensing, and less DNA damage. This study visualizes the plasticity of the cell cycle while providing a careful and quantitative examination of the impact of cyclin E on the cell cycle over many cell generations.

## Materials and Methods

### Cell culture and cell line construction

RPE1-hTERT cells were verified by STR profiling (ATCC) and confirmed to be mycoplasma negative. Cells were maintained in DMEM supplemented with 10% FBS, 2 mM L-glutamine, 1× pen/strep and incubated at 37°C in 5% $CO_2$. Cells were passaged with 1× trypsin before reaching confluency. RPE1-hTERT cells with the pINDUCER20-cyclin constructs were passaged every 3 d. To package lentivirus, pINDUCER20 (plasmid #44012; Addgene) with human cyclin E1, cyclin

D1, or cyclin A2 cDNAs were co-transfected with ΔNRF and VSVG plasmids (gifts from Dr. J Bear, UNC) into HEK293T cells with 50 μg/ml polyethylenimine (PEI)-Max (Aldrich Chemistry). RPE1-hTERT cells were transduced with the viral supernatant from the transfections in the presence of 8 μg/ml polybrene (Millipore) for 24 h, then selected with 500 μg/ml Geneticin (# 10131035; Gibco). Clones from the polyclonal population were isolated and screened for inducible expression of the respective protein (cyclin D1, cyclin E1, or cyclin A2).

Construction of reporter cell lines: PEI-Max was used to transfect PCNA-mTurquoise (Grant et al, 2018), DHB-mCherry (gift from S Spencer, University of Colorado-Boulder [Spencer et al, 2013]) into 293T cells using previous methods as stated above to package virus. The virus generated was then used to transduce RPE1-hTert cells already containing the pINDUCER20–cyclin E1 construct using the manufacturers' recommended protocol. Cells with stable integration of the plasmids were selected using 500 μg/ml G418 (Gibco). Colonies from single cells were selected by visual inspection for even expression of fluorescently-tagged proteins.

### Cloning

cDNAs were subcloned into the pINDUCER20 construct (Meerbrey et al, 2011) using either the Gateway cloning method (Life Technologies) or Gibson Assembly following protocols previously described (Matson et al, 2017; Grant et al, 2018). PCR fragments were amplified with Q5 polymerase (# M0491L; New England Biolabs, NEB). DNA fragments were isolated using the Qiaprep spin miniprep kit (QIAGEN). Plasmids were transformed into DH5α *Escherichia coli* strains. pENTR constructs were then combined with pINDUCER20 (plasmids#44012; Addgene). Plasmids were validated via sequencing (Eton Biosciences) for the desired insert using appropriate primers.

### Immunoblotting

Cells were lysed with CSK buffer (300 mM sucrose, 300 mM NaCl, 3 mM MgCl$_2$, and 10 mM PIPES, pH 7.0) containing 0.5% Triton X-100 supplemented with protease and phosphatase inhibitors (0.1 mM AEBSF, 1 μg/mL pepstatin A, 1 μg/mL leupeptin, 1 μg/mL aprotinin, 10 μg/ml phosvitin, 1 mM β-glycerol phosphate, and 1 mM Na-orthovanadate) for 20 min on ice. Lysates were centrifuged at 4°C for 10 min at maximum speed in a microcentrifuge. Supernatants were removed and lysates were diluted with SDS loading buffer (final: 1% SDS, 2.5% 2-mercaptoethanol, 0.1% bromophenol blue, 50 mM Tris, pH 6.8, and 10% glycerol) and boiled. Samples were separated on appropriate SDS–PAGE gels, and proteins transferred onto nitrocellulose (GE Healthcare) or polyvinylidene difluoride (PVDF) membranes (Thermo Fisher Scientific). After transfer, total protein was detected by staining with Ponceau S (Sigma-Aldrich). Membranes were blocked for 1 h at room temperature in 5% milk or 5% BSA in Tris-Buffered-Saline-0.1% Tween-20 (TBST). Membranes were then incubated in primary antibody for 16–18 h at 4°C with constant shaking in either 2.5% milk or 5% BSA in TBST with 0.01% sodium azide. After washing in TBST, membranes were incubated with horseradish peroxidase-conjugated secondary antibody in either 2.5% milk or 5% BSA in TBST for 1 h at room temperature and washed with TBST. Signals were detected using ECL Prime and a

ChemiDoc MP (Bio-Rad). This imaging system indicates any signals above the linear range of detection, and we did not include any images with signals outside that range. Antibodies used for immunoblotting were: cyclin E1 (Cat. no. 4129; Cell Signaling Technologies), cyclin D1 (Cat. no. sc753; Santa Cruz Biotechnology), cyclin A2 (Cat. no. 4656; Cell Signaling Technologies). Secondary antibodies used included: anti-rabbit IgG HRP-conjugated (1:10,000; Jackson ImmunoResearch) and goat anti-mouse IgG HRP-conjugated (1:10,000; Jackson ImmunoResearch).

### Immunoprecipitation

Cells were collected and frozen, then resuspended in Kischkel buffer (50 mM Tris, pH 8.0, 150 mM NaCl, 5 mM EDTA, and 1% Triton X-100) supplemented with 100 nM ATP and protease/phosphatase inhibitors as used for immunoblotting above for 20 min on ice. Cells were centrifuged at 4°C for 10 min at 16,000$g$ and the supernatants removed. Lysates were precleared for 45 min with magnetic beads (Dynabeads protein G, Cat. no. 10003D; Invitrogen) and nuclear digestion buffer (10 mM Hepes, pH 7.9, 10 mM KCl, 1.5 mM MgCl$_2$, 340 mM sucrose, and 0.1 mM glycerol) + 1 mM ATP + protease/phosphatase inhibitors (used above). After, quantification via Qubit or Bradford assay (Bradford, 1975), precleared lysates were adjusted to 1× high stringency IP buffer (Cat. no. 37510; Active Motif) with protease/phosphatase inhibitors, 0.01 mM ATP, and a non-ionic non-denaturing detergent (Cat. no. 37517; Active Motif). Lysates were then incubated with beads that had been pre-incubated with antibody in cold blocking buffer (1× PBS + protease/phosphatase inhibitors + ATP) for 6–8 h rotating at 4°C, then washed to remove unbound antibody. Lysates and antibody-bound beads were incubated with rotation at 4°C for 16–18 h. Unbound protein was removed the next day, beads were washed 3× with 5× high stringency IP buffer (diluted to 1×) supplemented with protease/phosphatase inhibitors + ATP and split 50% for immunoblotting and 50% for kinase assay. Antibodies used: cyclin E1 (Cat. no. 32-1500; Invitrogen) and cyclin E (Cat. no. 4129; Cell Signaling Technologies).

### Protein kinase assay

Protein–antibody–bead combinations were resuspended in kinase buffer (50 mM Tris, pH 7.5, 10 mM MgCl$_2$, and 0.01 mM ATP, + protease/phosphatase inhibitors). 2 μCi of [γ-$^{32}$P]ATP was added per sample, + 1 μg histone H1 protein (Cat. no. ab198676; Abcam). Samples were incubated for 30 min at 30°C. 20% 2-mercaptoethanol + sample buffer was added to each sample, boiled for 5 min, centrifuged 16,000$g$ for 30 s, then separated by SDS–PAGE, and the gel was stained with Coomassie blue, dried and exposed to a PhosphorImaging cassette (GE Healthcare). The phosphorylated histone H1 signal was quantified by subtracting the serum control from all experimental signals, and then all signals within one experiment were normalized to 1 based on activity in the control (endogenous cyclin E) sample.

### Flow cytometry

For analytical flow cytometry, EdU (Santa Cruz Biotechnology) was added to 10 μM 30 min before collection with trypsin. Cells were

permeabilized with 500 $\mu l$ cytoskeletal (CSK) buffer supplemented with 0.5% Triton X-100 and protease and phosphatase inhibitors as used above for immunoblotting, then fixed with PFA and subjected to antibody staining and EdU detection as described here dx.doi.org/10.17504/protocols.io.bba8iihw. Samples were analyzed on an Attune NxT flow cytometer (Life Technologies). Flow cytometry data were evaluated using FCS Express 7.0 software (De Novo). Cells were gated on FS(A) x SS(A) plots, and singlets were gated using DAPI(A) x DAPI(H) plots. Control samples were prepared as described previously (Matson et al, 2017). Control samples were treated the same as experimental samples by incubating with azide (either 647 or 488) and the appropriate secondary antibody. The following antibody/fluorophore combinations were used: (1) MCM (measuring origin licensing): Alexa 647-azide (Life Technologies), primary: Mcm2 (Cat. no. 610700; BD Biosciences), secondary: donkey anti-mouse-488 (Jackson ImmunoResearch), DAPI. (2) γ-H2AX (measuring DNA damage): Alexa 488-azide (Life Technologies), primary: γ-H2AX (Cat. no. 9718; Cell Signaling Technologies), secondary: donkey anti-rabbit 647 (Jackson ImmunoResearch), DAPI (Cat. no. D1306; Life Technologies).

## Doubling time

Doubling times were calculated by plating cells and counting cell numbers using a Luna II automated cell counter (Logos Biosystems) 72 h after plating. Each time course was repeated a minimum of three times. P = final # of cells, P0 = initial # of cells, PD = population doubling calculated as $\log_2(P/P0)$, and $D_t$ = population doubling time which was calculated as T (time)/PD. A minimum of three biological replicates were averaged for a final mean doubling time and graphed as $1/D_t$. GraphPad Prism and Excel were used to plot final graphs.

## Live cell imaging

RPE1-hTERT cells were plated as single cells in 24-well glass-bottom dishes (Cellvis) at a density of 20,000 cells/well. Cells were grown in FluoroBrite media (#A18967-01; Gibco) with 10% FBS, 2 mM L-glutamine, 1× pen/strep at 37°C in a humidified enclosure with 5% $CO_2$ (Okolab) and stimulated with 20 ng/ml doxycycline where indicated. Images were collected starting 4 h after plating, and cells were imaged for 72 h with images collected every 10 min using a Nikon Ti Eclipse inverted microscope (20× 0.75 NA dry objective lens) with the Nikon Perfect Focus system. Images were captured using an Andor Zyla 4.2 sCMOS detector with 12-bit resolution. All filter sets were from Chroma, CFP-436/20 nm; 455 nm; 480/40 nm (excitation; beam splitter; emission filter), YFP-500/20 nm; 515 nm; 535/30 nm; and mCherry-560/40 nm; 585 nm; 630/75 nm. Images were collected with the Nikon NIS-Elements AR software.

## Cell tracking

Cells were tracked using ImageJ software (https://imagej.nih.gov). Custom Python scripts (v3.7.1) in Jupyter Notebooks (v6.1.4) were used following tracking to perform single cell biosensor analyses. Individual cells were tracked and segmented in movies (time-lapse

experiments) using a previously described user-assisted approach (Grant et al, 2018).

## Immunofluorescence microscopy

Cells were fixed with 4% PFA for 15 min, permeabilized with 0.5% Triton X-100 for 30 min, then incubated with primary antibody (anti-53BP1, #NB100-304; Novus Biologics) at 4°C. The next day, cells were incubated with secondary anti-rabbit antibody conjugated to Alexa Fluor 488 for 1 h at room temperature. After washing, the cells were incubated with 1 $\mu g$/ml DAPI and imaged with appropriate filters. Slides were scanned with a Nikon Ti Eclipse inverted microscope (details in Live cell imaging section) with a 40× 0.95 NA objective. For each time point, 20 fields of view were manually selected for analysis yielding between 250 and 1,500 cells to be analyzed. Nuclei were detected based on DAPI signal using Python implementation of the Stardist segmentation algorithm (Weigert et al, 2020). 53BP1 nuclear bodies were detected using Difference of Gaussians image enhancement algorithm followed by a watershed segmentation algorithm using Scikit-image (van der Walt et al, 2014) in Python.

## Statistical analysis

Bar graphs represent means, and error bars indicate SEM, unless otherwise noted. The number and type of replicates are indicated in the figure legends. Significance tests were performed using a one-way ANOVA test, as indicated in the figure legends, unless otherwise specified. Statistical significance is indicated as asterisks in figures: $*P \leq 0.05$, $**P \leq 0.05$, $***P \leq 0.005$ and $****P \leq 0.0005$. GraphPad Prism v.8.0 and Python were used for statistical analysis.

## RNA extraction, library preparation, and RNA sequencing

RNA was extracted from RPE1-hTERT cells using the Quick-RNA Miniprep Kit (Zymo Research) with chloroform extraction. Libraries for each sample were prepared, at the same time, using KAPA Hyperprep mRNA Library Kit (Roche) and Illumina adapters (New England Biolabs) starting with 1 $\mu g$ RNA for each sample. Pooled libraries were sequenced with a Nextseq 500 high output 75 cycle kit for single-end reads.

To analyze RNA sequencing data, Kallisto (v0.44.0) was used to quantify transcript abundance against a reference human genome (hg38) followed by generation of transcripts per million tables. Kallisto output was imported into DESeq2 (v1.24.0) to determine differential gene expression between time points. Ranked lists of genes with differential expression between time points based on $\log_2$ fold change were used for gene set enrichment analysis. Pairwise comparisons were performed using detected genes with transcripts per million of at least five. Cell cycle gene sets were derived from RNA sequencing (Dominguez et al, 2016) and in total 931 genes overlapped with this dataset. Periodic "seeds" 1–7 from Dominguez et al (2016) were used to investigate expression changes in this study. Breast cancer (BRCA) data from Tumor Cancer Genome Atlas Data were downloaded from the Cancer Bioportal Repository (Gao et al, 2013). Copy number variation and gene expression values (RSEM + 1) were used from the published datasets. Statistical

significance tests are denoted in figure legends with *P*-value adjustment when multiple comparisons were made.

## Data Availability

The data are included in the figures and supplements. RNA-Seq data have been deposited in the Gene Expression Omnibus database (accession no. GSE171845).

### Supplementary data

Figs S1–S6 and Table S1 are provided.

## Supplementary Information

## Acknowledgements

We thank Sidra Qayyum, Megan Justice, Cyrus Vaziri, Michael Emanuele, Lee Graves, and Wayne Stallaert for technical assistance. We thank Jeffrey Jones for lab management. We thank Samuel Wolff and Jeremy E Purvis for expertise in live cell fluorescence microscopy. The results shown here are in whole or part based upon data generated by the TCGA Research Network: https://www.cancer.gov/tcga. The project was supported by grants from the National Institutes of Health: R01GM083024, R01GM102413, and R35GM141833 to JG Cook; UNC-CH Cancer Control Education Program 5T32CA057726-28 to A Walens, UNC start-up funds to D Dominguez, the Chan Zuckerberg Initiative DAF (an advised fund of Silicon Valley Community Foundation 2020-225716) to KM Kedziora, and Burroughs Wellcome Fund Graduate Diversity Enrichment Program award 1020278 to JC Limas. BL Mouery was supported by NIH/NIGMS T32GM135128, and JC Limas was supported by an HHMI Gilliam Fellowship for Advanced Study (GT10886), and NIH/NIGMS awards to UNC: R25GM055336 and T32GM007040. The UNC Hooker Imaging Core and Flow Cytometry Core Facility are supported in part by P30 CA016086 Cancer Center Core Support Grant to the UNC Lineberger Comprehensive Cancer Center. This research was also supported in part by the North Carolina Biotech Center Institutional Support Grant 2017-IDG-1025 and by the National Institutes of Health 1UM2AI30836-01. The content is solely the responsibility of the authors and does not necessarily represent the official views of the National Institutes of Health.

## Author Contributions

JC Limas: conceptualization, formal analysis, supervision, funding acquisition, validation, investigation, project administration, and writing—original draft, review, and editing.
AN Littlejohn: investigation and writing—original draft.
AM House: investigation.
KM Kedziora: conceptualization, formal analysis, funding acquisition, validation, investigation, and writing—original draft, review, and editing.
BL Mouery: investigation.
B Ma: investigation.
D Fleifel: investigation.
A Walens: formal analysis and writing—original draft.
MM Aleman: formal analysis, validation, investigation, and writing—original draft, review, and editing.
D Dominguez: formal analysis, validation, investigation, and writing—original draft, review, and editing.
JG Cook: conceptualization, formal analysis, supervision, funding acquisition, validation, investigation, project administration, and writing—original draft, review, and editing.

**Conflict of Interest Statement**

The authors declare that they have no conflict of interest.

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
