## [Reviewer comments · Life Science Alliance]

Life Science Alliance

Quantitative Profiling of Adaptation to Cyclin E Overproduction

Juanita Limas, Amiee Littlejohn, Amy House, Katarzyna Kedziora, Brandon Mouery, Boyang Ma, Dalia Fleifel, Andrea Walens, Maria Aleman, Daniel Dominguez, and Jeanette Cook

DOI: <https://doi.org/10.26508/lsa.202201378>

Corresponding author(s): Jeanette Cook, University of North Carolina at Chapel Hill

Review Timeline:	Submission Date:	2022-01-21
	Editorial Decision:	2022-01-24
	Revision Received:	2022-01-28
	Accepted:	2022-01-31

Transaction Report:

Please note that the manuscript was reviewed at *Review Commons* and these reports were taken into account in the decision-making process at *Life Science Alliance*.

Review #1

Cyclin E is commonly overexpressed in cancer cells and many previous studies have documented the various genotoxic consequences of deregulated cyclin E activity, which include G1 acceleration, premature S-phase entry, delayed S-phase progression and replication stress, impaired break induced DNA repair, and DNA damage. However, while these outcomes are deleterious to proliferation, cyclin E overexpression is commonly found in cancers, raising the possibility that cancer cells may adapt to chronic cyclin E overexpression. This manuscript seeks to address these issues by using state of the art approaches to examine the acute and chronic effects of ectopic cyclin E expression in RPE-TERT cells. The authors use a dox-inducible system to conditionally express cyclin E, or alternatively express cyclins D and A as controls. Shortly after induction, cyclin E was uniquely found to greatly shorten G1 and drive early S entry. The cyclin E overexpressors prematurely entered S-phase with under-licensed origins and up to 10x less MCM loading in early S phase cells. Licensed yet dormant origins have been shown to protect against replication stress in other contexts and the authors show here that cyclin E overexpressors are hypersensitive to gemcitabine.

The rest of the paper is largely focused on the question of how these cells continue to proliferate despite under licensed origins. They find that the cyclin E cells undergo a proliferative crisis from which they emerge in an adapted state. While the obvious possibility was that the adapted cells simply downregulated cyclin E-CDK2 activity in some way, this was not the case. Instead, the adapted cells acquired an intermediate licensing state despite high cyclin E-CDK2 activity, and this state persisted after cyclin E was turned off, suggesting an adaptation that extended beyond a simply acute and/or ongoing response to CDK2 activity, and that may involve a prolonged DNA damage response.

The authors use CDK1/2 biosensors and single cell analyses to study the cellular response to cyclin E overexpression over time. Perhaps not surprisingly, this revealed a heterogenous response with either continued proliferation or arrest in a few different cell cycle states. These results are reminiscent of early studies with these reporters that described a CDK2 dependent commitment to cell cycle re-entry that is established in the prior mitosis. They used RNA-Seq to characterize the transcriptional state of cyclin E-adapted cells over time and note two genes, p21 and HORMAD1, that differed in acute versus chronic cyclin E overexpressors. To generalize these results to cancers, they assessed TCGA databases and found a correlation between cyclin E and HORMAD1 overexpression. HORMAD1 is a testis-cancer antigen with a role in mitosis, and they suggest that HORMAD1 overexpression helps cancers with cyclin E overexpression tolerate cyclin E-generated replication stress.

The causes and consequences of deregulated cyclin E have been well- studied, particularly in the areas of replication stress and DNA damage. However, many underlying mechanisms are still not understood, and this paper is a comprehensive demonstration of applying new technologies to study important and unresolved cell cycle problems. In particular, the dynamic studies of adaptation and the single-cell analyses are very well done. In my opinion, the main question is, not whether this is excellent work but rather "where does this paper make the most significant new insights?" That is, it is sometimes more a matter of providing very clear demonstrations and increased knowledge, rather than fundamental new models or insights. One example is the elegant demonstration of origin "underlicensing" in the background of the known decrease in MCM loading caused by excess cyclin E. Another is the careful demonstration of the distinction between the acute and chronic consequences of cyclin E expression. Others have described these cellular responses to cyclin E overexpression, but this paper does so very nicely. The manuscript adds important granularity to these questions.

In my opinion, the most novel parts of the paper are: 1) the proliferative crisis in response to excess cyclin E activity, 2) the persistent "intermediate licensed state" after cyclin E is repressed, and 3) the potential role of HORMAD1. In general, the manuscript has much stronger descriptive data around these three areas than mechanistic insights, which is not a criticism per se but more a reflection of the methods.

A few specific points:

1. The identification of HORMAD1 as a potential cyclin E-tolerizing mechanism is very interesting. However, while the paper validates the correlation between cyclin E and HORMAD1 expression, it only speculates that HORMAD1's functions may help to resolve cyclin E-induced replication stress. It would be more compelling if the authors could directly test this hypothesis. That is, does HORMAD1 over- or under- expression modify cyclin E-induced replication stress sensitivity or cellular capacity to tolerate this stress? This question seems within the scope of the paper.

2. While the defect in MCM loading is nicely demonstrated it has not been shown to be causal. It is also not clear if this reflects a direct consequence of cyclin E-CDK2 activity or just a shorter G1 phase. The discussion favors the latter, but it would be good to have a better sense if this is the case. It is also unclear to what extent the reduced loading directly accounts for the sensitivity of cyclin E-overexpressors to replication stress. This can be hard given the many homeostatic mechanisms that regulate MCM loading and origin licensing, but something that shows its direct role would help strengthen this idea.

3. The authors premise the paper on the question of how tumor cells tolerate chronic cyclin E overexpression. However, many cancer cells with high cyclin E expression have other mutations that impinge upon CDK2 activity, such as p53 loss, chk1 mutations, CDC25 gain, etc. One of the advantages of RPE cells is that they have intact p53-21 and normal S-phase checkpoint controls. However, this is not typically the case for cancer cells. Are the acute and chronic responses to cyclin E modified by these types of mutations? That is, how well does the RPE system reflect what is happening in cancer cell? For example, the increased CDK2 activity seems modest compared with the amount of cyclin E overexpression (Fig. 1). While this could reflect limited CDK2 available for cyclin E to bind, it could also be due to p53-p21 activation.

4. Are the complex adaptive responses described in RPE cells also found in other cell types? If not, why is there cell type specificity and how might this impact the cancer relevance?

****Referees cross-commenting****

All 3 reviews are quite concordant- major difference is in how

issues are perceived. The two major things in all 3 reviews are 1) lack of novel mechanistic insights, and 2) descriptive and correlative parts of the paper, especially HORMAD1. Main questions to me are whether these issues will be addressed by any (or all) of the specific items in each review and do they need to be for publication? The latter depends on the journal I suppose. Otherwise the paper is pretty solid.

Review #2

This manuscript addresses an interesting question: what are the consequences of cyclin E overexpression in cancer cells. As a model system the authors use RPE1-TERT cells that inducibly overexpress cyclin E. The authors show that the cells go through 3 phases: the first phase lasts about 3 days and is associated with increased proliferation; the second phase lasts about 20 days and is associated with decreased proliferation and the third phase is associated with normal proliferation. The authors describe well these 3 phases in regard to cell proliferation, but then provide very few mechanistic insights. One is left somewhat disappointed after Fig 4. Certainly the manuscript can be published, but the emphasis on HORMAD1 should be downplayed and HORMAD1 should not even be mentioned in the abstract, unless the authors can provide something more than correlative data.

****Specific Points****

1. Figs 1 and 2 refer to the first phase of increased cell proliferation.

2. Fig. 2E shows a significant decrease in bound MCM after cyclin E overexpression. But in Fig 3E, the decrease in bound MCM is much less. In Fig 3F, the decrease is very strong. What is the explanation for these different results? Which one is true?

3. Fig. 3 describes all three phases, but there are no molecular data for the second phase. The authors examine: day 0 (no induction of cyclin E), day 2 (the first phase) and day 30+ (the third phase). Unfortunately, the experiments need to be repeated to include the second phase.

4. Fig 3B shows a "small" decrease in cyclin E levels at the third phase. However, I think that there is a large decrease of cyclin E levels, which would have become apparent if the blot was not overexposed. This is consistent with the data shown in Fig 5A, which shows a large decrease in cyclin E mRNA levels in phase 3.

5. Fig. 4 no concerns, but descriptive.

6. Fig. 5 shows very few changes in gene transcription between the control vs acute and acute vs adapted phases. There are other genes with greater L2FC than HORMAD1. Which ones are these? Do HORMAD1 levels and the levels of the other genes with high L2FC decrease in the adapted cells? There are no experiments addressing whether the increase in HORMAD1 levels have any functional significance. Are they related to the three phases of cell proliferation described above? If so, how?

7. What is the take-home message of this study? It seems very preliminary.

****Referees cross-commenting****

The other reviewers also state that the study is descriptive and that the link to HORMAD1 is merely a correlation.

Review #3

It has been known for a long time that overexpression of cyclin E shortens G1 and induces cells to enter S phase with a reduced number of licensed replication origins, resulting in replicative stress. This effect is of interest because cyclin E is overexpressed in some cancer cells. In the current paper, Limas et al investigate overexpression of cyclin E and make several observations that provide a clearer understanding of how it affects the cell cycle. They show that whilst cyclin E overexpression in immortalised RPE1 cells initially results in reduced cell proliferation, over a period of weeks the cells adapt and proliferation rates recover. Single-cell analysis shows that the acute reduction in proliferation is, as expected, associated with a shortened G1 and a lengthened S phase; surprisingly the adapted cells recover a more normal cell cycle pattern. The adapted cells have not suppressed cyclin production with associated kinase activity being still as high as in the acutely affected cells. The authors argue that this adaptation is not due to the expansion of rare mutant clones but is a non-genetic response of a large proportion of the population as it happens over a relatively short period to relatively small cell cultures. The molecular explanation for this adaptation remains unclear, but transcriptomic analysis shows that it is associated with expression of the HORMAD1 gene. The authors also show a good correlation between cyclin E and HORMAD1 expression in different cancer cells. The authors also provide arguments that the effect of underlicensing is an indirect effect of cyclin E mediated by it pushing cells prematurely into S phase, where other activities, such as cyclin A-CDK2, lead to suppression of the licensing process by known mechanisms.

The experiments are generally of a high quality and provide good support for the arguments being made. I have a few minor criticisms and suggestions for improvement, but nothing major in this regard (see minor points below).

****Minor points****

1. 1st paragraph of the Introduction: 'In G1 phase, cyclin D activates CDK4 and CDK6 to stimulate E2F-dependent transcription of a cohort of genes necessary for S phase entry.' I know this is just a summary to get readers up to speed, but the description should be consistent with the known role of cyclin E in activating E2F. One possible re-writing would be to say: 'In G1 phase, cyclin D activates CDK4 and CDK6 to initiate the stimulation of E2F-dependent...'

2. The Mendez and Stillman 2000 citation is a bit of a strange choice alongside Evrin 2009 (note spelling throughout, not Ervin) and Remus and Diffley, 2009. Gillespie et al 2001 would be a better citation for the 4 licensing activities.

3. 1st paragraph of Results: 'Immunoblotting for each G1-S cyclin showed no effect of overproducing one cyclin on expression of another cyclin'. That doesn't appear to be true: In Fig 1 A overexpression of cyclin A causes increased expression of cyclin E - that's exactly what I would expect if cyclin A causes maximal Rb phosphorylation and hence maximal activation of E2F. The text should be amended to describe the figure more accurately.

4. Top of page 5: I can't see the data in the supplemental figure 1A-B about doubling time as is stated in the main text. To use doubling time to estimate phase length depends on there being no change to the rate of cell death - was that checked?

5. Overproducing cyclin E only increases its kinase activity ~2 fold (Fig 1E), but expression levels of cyclin E are increased by ~10 fold (Fig 1B). I think that difference would be worth pointing out in the main text, as it is consistent with compensatory regulation of kinase activity for example by p21.

6. Fig 2C is not quite as clear as it could be (for example, Supplemental Figure 1C looks better). Either replace it with cleaner data or perhaps cut the y axis at 100.

7. Figure 2E - how was this data normalised? I'd expect the exact MCM signal to vary between experimental replicates, and I presume the control values were normalised to 100 - if so, the figure legend should state this.

8. Figure 2E - I think it would be worth bringing the charts showing mean MCM loading for cyc D and cycA from Supplemental Figure 1D and 1F into main Figure 2. The difference in effect between the three cyclins is less extreme than I might have imagined. I think the text overstates the effect of cyclin E: 'By this analysis, cyclin E overproduction reduced licensing in early S by as much as ten-fold relative to control cells.' It would be fairer to say 'By this analysis, cyclin E overproduction reduced licensing in early S by approximately 3-fold.'

9. Figure 2F: the number of 53BP1 nuclear bodies would be a more direct way to demonstrate the effect of under-licensing than gamma-H2AX foci.

10. Figure 3A and Supplemental Figure 3A: the unit on the y axis data is not adequately described in the legend - I'm guessing this is 1/hours, even though the x axis is in days. It might be more intuitive to use in the same units in both (hours or days). Was there any change in cell death during the experiment, as that would influence calculations of the doubling time?

11. Figure 3A, inset. The legend doesn't really describe this experiment properly; the main text indicates that the '-' samples are from cells initially treated with dox and then had the dox withdrawn, but for how long is not clear.

12. Why is the 30+ day cyclin E-induced decrease in MCM2 loading so much more pronounced in Figure 3F (looks ~80% reduction after 34 days) than 3E (looks ~20% reduction after 30+ days)?

13. Figure 4A - the large 53BP1 foci are called 'Nuclear Bodies' (Lukas et al, 2011).

Whilst the paper clearly advances our understanding of the effect of cyclin E overexpression, the major weakness of the work is that no really new mechanistic understanding is given. The adaptation is an interesting effect and could well represent events occurring in the development of cancers associated with cyclin E overexpression, but it remains rather mysterious. The associated expression of *HORMAD1* is intriguing and worthy of further investigation, but at present it remains simply a correlation. *HORMAD1* has been linked to increased tolerance of DNA breaks so it would have been to know, for example, if knocking down *HORMAD1* reduces the adaptation of cells to cyclin E overexpression.

****Referees cross-commenting****

To be clear, I think the *HORMAD1* data, even though correlative, is interesting and is appropriately described by the text. If the authors want to increase the impact of the work, they could do experiments that all 3 reviewers have suggested to show its functional importance; however, if the authors don't want to spend the time doing this and are content with having less impact, I think the manuscript is publishable in approximately its current form, so long as the other relatively minor issues are addressed.

Review #4

In this study from the JC Cook lab, Limas et al investigate how non-cancerous, immortalized human epithelial cells (RPE1-hTERT) may adapt to the chronic overexpression of cyclin E1. They analyze how cells behave in terms of proliferation rate and origin licensing capacity, i.e. the recruitment of MCM proteins to DNA in order to initiate DNA replication. Following a short burst of rapid proliferation, RPE1-hTERT cells expressing high levels of *CycE* undergo a proliferative crisis characterized by incomplete origin licensing and very slow cell cycles. Some cells eventually adapt to the high *CycE* levels and end up proliferating slightly faster than control cells, while origin licensing is partially recovered. The authors perform RNA sequencing at different time points across the process, trying to identify a signature of transcriptional changes that could explain the adaptation. A promising target is found (*HORMAD1*), and computational analyses reveal that the levels of *HORMAD1* also correlate with those of *CycE1* in basal-like breast cancers.

****Major points:****

1. The study is well designed and the experiments seem quite logical as a first attempt to understand the -presumably complex- mechanisms underlying the adaptation of normal cells to grow in the presence of high levels of *CycE*. The "adaptation phenotypes" are well described in Figures 1-4 (even if some observations are confirmatory of previous studies, some of them published in the 1990s and 2000s). However, in my opinion the study somehow falls short in terms of identifying the possible adaptation mechanism(s). The transcriptomics approach shown in Figure 5B reveals global changes in G1-expressed genes that could be both cause or consequence of the adaptation, as properly acknowledged by the authors in the text. The analyses shown in Figure 5C and 5D reveal few changes that pass the statistical filters and make clear sense as a whole. This leaves *HORMAD1* as the most promising candidate, but its role is not validated (see below).

2. While I agree that some of the authors' observations are not consistent with genetic mutations driving adaptation, several parameters are being monitored simultaneously (duration of G1, origin licensing, rate of cell division) and it is conceivable that genetic mutations may drive or at least influence some of them, particularly the ability to overcome

the proliferation crisis. In this regard, the adaptation to high CycE levels has parallelisms with the well-studied cell immortalization process.

3. The identification of *HORMAD1* gene upregulation is interesting because this gene is preferentially meiotic, and the authors find other instances in which *HORMAD1* expression correlates with the levels of *CCNE1* (e.g. breast cancer). Indeed, a large part of the discussion is focused on *HORMAD1*. However, the causal role of *HORMAD1* remains speculative at this point and it would require some fundamental molecular characterization, e.g. could adaptation occur in its absence?, etc.

****Minor points:****

1. Figure 1A. The text argues that ectopic expression of one Cyclin does not influence the levels of the others. While this would not be a critical issue, I believe some fluctuations may be obscured by the long-exposure blots shown (particularly *cycE* in the left column (after cyclin D o/e), and *cycD* in the right column (after cyclin A o/e)).

2. Figure 2F. The conclusion that cells overexpressing *cycE* are hypersensitive to gemcitabine seems derived from the statistical significance of the difference between lanes 2 and 4. However, the fold-increase in *γH2AX* caused by GEM in control cells (lanes 1-3) is similar, or probably higher, than in *cycE*-overexpressing cells. In order to demonstrate defects in dormant origin activation, the use of stretched DNA fibers would be a more direct approach.

3. In the Introduction (p.4, l.4) the authors state that "some tumor-derived cell lines with high CycE proliferate without experiencing similar levels of replication stress or impaired origin licensing (Ekholm-Reed et al, 2004,...). However, the study of Ekholm-Reed et al (2004) shows precisely that even carcinoma cells display impaired origin licensing upon *cycE* overexpression.

The study is pertinent as part of a long-time effort by several laboratories to understand the effects of abnormal *cycE* levels in genomic stability, which was started by the pioneering work of Steve Reed and others in the mid 1990s. The topic is very interesting and should be appealing to scientists interested in cell cycle, CDKs, DNA replication and cell transformation by oncogenes.

As mentioned before, some of the observations about the cellular response to *cycE* overexpression recapitulate previous findings (e.g. shorter G1 phase, origin under-licensing, replicative stress), even if some technical approaches are new and the cell lines are different. The most original part is the transcriptional profiling of the adaptation process, which still sounds preliminary. I believe the manuscript would greatly improve if the participation of *HORMAD1* could be demonstrated directly.

1. General Statements [optional]

This section is optional. Insert here any general statements you wish to make about the goal of the study or about the reviews.

We thank the editors and four reviewers for their time and thoughtful consideration of our study. We are gratified that the reviewers appreciated the sophistication of the techniques, the “high quality” of the experiments, and the importance of the general topic. Reviewers commented on the “state of the art approaches”, “comprehensive demonstration of applying new technologies to study important and unresolved cell cycle problems, “and the “elegant demonstration of origin ‘underlicensing’”. We have now provided a full revision for editorial consideration.

The theme of the reviewers’ first critiques regarding significance is that the study is largely descriptive rather than deeply mechanistic. To some extent we agree, but we also assert that careful and quantitative descriptive studies have always been valuable. Although others have analyzed cells responding to cell cycle-inhibitory drugs or compared a final resistant population to the starting population, to the best of our knowledge no other study achieves such a deep analysis of cell populations throughout the entire process of adaptation. We document the variety of individual cell fates during the proliferative crises

(reviewer 1 notes this is “important granularity”) and provide evidence that the adaptation to the stress of cyclin E overproduction includes non-genetic mechanisms. Reviewers requested functional data on the mechanisms of adaptation, particularly the potential role for HORMAD1. Given that the mechanisms of adaptation involve multiple changes, we predict that testing and defining the functions of those mechanisms with respect to the adaptation process will be another full project. For that reason, we agree that this finding is preliminary and have removed HORMAD1 discussion from this manuscript. These changes involved revisions to the original abstract, figure 6 and supplemental figure S6, and associated results and discussion sections.

Moreover, we have demonstrated a second mechanism by which mammalian cyclin-dependent kinase can inhibit origin licensing. The paradigm in the field is that CDKs inactivate origin licensing through phosphorylating and directly inhibiting MCM loading proteins. This paradigm is founded on the relationship between origin licensing proteins and S phase CDKs in yeasts. We now show that the CDK most responsible for inducing origin firing in mammalian cells, cyclin E/CDK2, inhibits licensing *indirectly* by inducing premature S phase entry which leaves insufficient time in G1 for normal licensing. We discuss this finding and supporting evidence from the literature that strongly argues against cyclin E/CDK2 as a direct licensing inhibitor

(page 13-14). Our study thus explores a new concept for the relationship between CDK activity and origin licensing in mammalian cells. We anticipate that this work is not only a meaningful contribution to understanding cell cycle plasticity, but it can also be the foundation and inspiration for readers to generate new hypotheses.

The first revision was evaluated by 3 of the reviewers with mixed reactions (we assume Rev 1-3 in the second round are also Rev 1-3 from the first round). The remaining concerns relate to significance and the lack of mechanism plus a new concern from Reviewer 2 that the phenotypes may not be related to adaptation per se. Reviewer 1 is satisfied with the first revision. We have responded to these second reviews below our responses to the first comments and provide a second revision with additional changes. Per a reviewer's request, we have included the full text of the reviewers comments and used quotations and formatting to identify our responses and the first and **second** reviews.

We are confident that the additionally revised manuscript and figures are improved and we anticipate that your readers will appreciate this addition to the field. Thank you for your consideration.

This section is mandatory. Please insert a point-by-point reply describing the revisions that were already carried out and included in the transferred manuscript.

Reviewer 1 - First review

Evidence, reproducibility and clarity (Required): Cyclin E is commonly overexpressed in cancer cells and many previous studies have documented the various genotoxic consequences of deregulated cyclin E activity, which include G1 acceleration, premature S-phase entry, delayed S-phase progression and replication stress, impaired break induced DNA repair, and DNA damage. However, while these outcomes are deleterious to proliferation, cyclin E overexpression is commonly found in cancers, raising the possibility that cancer cells may adapt to chronic cyclin E overexpression. This manuscript seeks to address these issues by using state of the art approaches to examine the acute and chronic effects of ectopic cyclin E expression in RPE-TERT cells. The authors use a dox-inducible system to conditionally express cyclin E, or alternatively express cyclins D and A as controls. Shortly after induction, cyclin E was uniquely found to greatly shorten G1 and drive early S entry. The cyclin E overexpressors prematurely entered S-phase with under-licensed origins and up to 10x less MCM loading in early S phase cells. Licensed yet dormant origins have been shown to protect against replication stress in other contexts and the authors show here that cyclin E overexpressors are hypersensitive to gemcitabine.

The rest of the paper is largely focused on the question of how these cells continue to proliferate despite under licensed origins. They find that the cyclin E cells undergo a proliferative crisis from which they emerge in an adapted state. While the obvious possibility was that the adapted

cells simply downregulated cyclin E-CDK2 activity in some way, this was not the case. Instead, the adapted cells acquired an intermediate licensing state despite high cyclin E-CDK2 activity, and this state persisted after cyclin E was turned off, suggesting an adaptation that extended beyond a simply acute and/or ongoing response to CDK2 activity, and that may involve a prolonged DNA damage response.

The authors use CDK1/2 biosensors and single cell analyses to study the cellular response to cyclin E overexpression over time. Perhaps not surprisingly, this revealed a heterogenous response with either continued proliferation or arrest in a few different cell cycle states. These results are reminiscent of early studies with these reporters that described a CDK2 dependent commitment to cell cycle re-entry that is established in the prior mitosis. They used RNA-Seq to characterize the transcriptional state of cyclin E-adapted cells over time and note two genes, p21 and HORMAD1, that differed in acute versus chronic cyclin E overexpressors. To generalize these results to cancers, they assessed TCGA databases and found a correlation between cyclin E and HORMAD1 overexpression. HORMAD1 is a testis-cancer antigen with a role in mitosis, and they suggest that HORMAD1 overexpression helps cancers with cyclin E overexpression tolerate cyclin E-generated replication stress.

The causes and consequences of deregulated cyclin E have been well- studied, particularly in the areas of replication stress and DNA damage. However, many underlying mechanisms are still not understood, and this paper is a comprehensive demonstration of applying new technologies to study important and unresolved cell cycle problems. In particular, the dynamic studies of adaptation and the single-cell analyses are very well done. In my opinion, the main question is, not whether this is excellent work but rather "where does this paper make the most significant new insights?" That is, it is sometimes more a matter of providing very clear demonstrations and increased knowledge, rather than fundamental new models or insights. One example is the elegant demonstration of origin "underlicensing" in the background of the known decrease in MCM loading caused by excess cyclin E. Another is the careful demonstration of the distinction between the acute and chronic consequences of cyclin E expression. Others have described these cellular responses to cyclin E overexpression, but this paper does so very nicely. The manuscript adds important granularity to these questions.

In my opinion, the most novel parts of the paper are: 1) the proliferative crisis in response to excess cyclin E activity, 2) the persistent "intermediate licensed state" after cyclin E is repressed, and 3) the potential role of HORMAD1. In general, the manuscript has much stronger descriptive data around these three areas than mechanistic insights, which is not a criticism per se but more a reflection of the methods.

1. "The identification of HORMAD1 as a potential cyclin E-tolerizing mechanism is very interesting. However, while the paper validates the correlation between cyclin E and HORMAD1 expression, it only speculates that HORMAD1's functions may help to resolve cyclin E-induced replication stress. It would be more compelling if the authors could directly test this hypothesis. That is, does HORMAD1 over- or under- expression

modify cyclin E-induced replication stress sensitivity or cellular capacity to tolerate this stress? This question seems within the scope of the paper.” **Reviewer 1** suggests an improvement to the study by testing the effects of HORMAD1 manipulation (#1). Other reviewers made similar comments.

Although the fold-change in HORMAD1 mRNA detected by mRNA sequencing was the largest among all the protein-coding genes (aside from cyclin E), the absolute expression level was actually low. This limitation precludes many of the functional assays we could attempt, so we have taken Reviewer 2’s suggestion to refocus the paper away from this interesting, but admittedly preliminary result. These changes involved revisions to the abstract, figures 6 and supplemental figure 6.

2. *“While the defect in MCM loading is nicely demonstrated it has not been shown to be causal. It is also not clear if this reflects a direct consequence of cyclin E-CDK2 activity or just a shorter G1 phase. The discussion favors the latter, but it would be good to have a better sense if this is the case. It is also unclear to what extent the reduced loading directly accounts for the sensitivity of cyclin E-overexpressors to replication stress. This can be hard given the many homeostatic mechanisms that regulate MCM loading and origin licensing, but something that shows its direct role would help strengthen this idea.”*

In the revised text, we have emphasized that prior work indeed established that genome damage in the form of γ -H2AX increases and replication stress hypersensitivity is caused by inhibiting origin licensing and provided additional citations (page 6). We can of course, not rule out any additional non-licensing contributions from cyclin E overproduction in this study, and any manipulations to divorce G1 length from CDK2 activity would be exceedingly difficult (and perhaps biologically impossible).

3. *“The authors premise the paper on the question of how tumor cells tolerate chronic cyclin E overexpression. However, many cancer cells with high cyclin E expression have other mutations that impinge upon CDK2 activity, such as p53 loss, chk1 mutations, CDC25 gain, etc. One of the advantages of RPE cells is that they have intact p53-p21 and normal S-phase checkpoint controls. However, this is not typically the case for cancer cells. Are the acute and chronic responses to cyclin E modified by these types of mutations? That is, how well does the RPE system reflect what is happening in cancer cell? For example, the increased CDK2 activity seems modest compared with the amount of cyclin E overexpression (Fig. 1). While this could reflect limited CDK2 available for cyclin E to bind, it could also be due to p53-p21 activation.”*

Although framed as a limitation, we suggest that the isolation of one genetic change is a strength because we measured the effects of that one change over time. Although we did not add any new experiments to address this point because it would be beyond the scope, we have included additional discussion of these complexities (pages 4, introduction and 14, discussion).

4. "Are the complex adaptive responses described in RPE cells also found in other cell types? If not, why is there cell type specificity and how might this impact the cancer relevance?"

Although we agree that it would be very interesting to determine if the adaptive responses we observe are general, the time and resources needed to replicate this study in other cell lines is beyond the scope of the study. Nonetheless, the DNA damage response and genome instability from acute cyclin E overproduction were reported in both Minella et al 2002 (phospho-p53) and Teixeira et al 2015 (chromosome-instability). The adaptation itself is unique to our current study however. (See also our response to Reviewer 1 point #3)

****Referees cross-commenting****

"All 3 reviews are quite concordant- major difference is in how issues are perceived. The two major things in all 3 reviews are 1) lack of novel mechanistic insights, and 2) descriptive and correlative parts of the paper, especially HORMAD1. Main questions to me are whether these issues will be addressed by any (or all) of the specific items in each review and do they need to be for publication? The latter depends on the journal I suppose. Otherwise the paper is pretty solid."

Reviewer 1 - Second review. In response to the first revision, Reviewer 1 is satisfied and writes "I think this is a very nice paper and is appropriate for publication."

Reviewer 2 - First review (Evidence, reproducibility and clarity (Required): *"This manuscript addresses an interesting question: what are the consequences of cyclin E overexpression in cancer cells. As a model system the authors use RPE1-TERT cells that inducibly overexpress cyclin E. The authors show that the cells go through 3 phases: the first phase lasts about 3 days and is associated with increased proliferation; the second phase lasts about 20 days and is associated with decreased proliferation and the third phase is associated with normal proliferation. The authors describe well these 3 phases in regard to cell proliferation, but then provide very few mechanistic insights. One is left somewhat disappointed after Fig 4. Certainly the manuscript can be published, but the emphasis on HORMAD1 should be downplayed and HORMAD1 should not even be mentioned in the abstract, unless the authors can provide something more than correlative data."*

1. "Figs 1 and 2 refer to the first phase of increased cell proliferation."
(no question or critique)
2. "Fig. 2E shows a significant decrease in bound MCM after cyclin E overexpression. But in Fig 3E, the decrease in bound MCM is much less. In Fig 3F, the decrease is very strong. What is the explanation for these different results? Which one is true?" Reviewer 2 points to the difference in the magnitude of underlicensing induced by different examples in Figure 2E and Figure 3E.

The absolute extent of underlicensing can vary within biological replicates performed at different times, but the underlicensing itself is consistent. Figure 2E is the aggregate data from 20 independent experiments, and the mean difference is ~2.5-fold whereas the example in Figure 3E is still underlicensed, but less so. We have replaced Figure 2C and 2D with a less dramatic example (which was admittedly the of the most dramatic examples) and revised the text to make it more clear that there are ranges of underlicensing.

3. *"Fig. 3 describes all three phases, but there are no molecular data for the second phase. The authors examine: day 0 (no induction of cyclin E), day 2 (the first phase) and day 30+ (the third phase). Unfortunately, the experiments need to be repeated to include the second phase."*

We are somewhat perplexed because there are molecular data in Figure 4A across the entirety of an adaptation plus analysis of CDK activity at each of the four critical time points. For this reason, we do not agree that we must repeat the adaptation because all of the molecular data in Figure 3 is in Figure 4, just a different method for measuring CDK activity. Nonetheless, we have included additional immunoblots for cyclin E and p27 in Figure 4B.

4. *"Fig 3B shows a "small" decrease in cyclin E levels at the third phase. However, I think that there is a large decrease of cyclin E levels, which would have become apparent if the blot was not overexposed. This is consistent with the data shown in Fig 5A, which shows a large decrease in cyclin E mRNA levels in phase 3."*

We certainly agree that cyclin E protein is reduced as we mentioned in the text, although the magnitude of that effect varies among independent adaptations. We chose a long exposure of the immunoprecipitation to detect endogenous cyclin E in lane 6. Moreover, these images are not scanned autoradiography films, but rather, captured on an imaging system that indicates saturation; no exposures in the paper included saturated signals. We have made this exposure choice explicit in the methods (page 16-17), and we provide lighter exposures in the new Supplemental Figure S6.

5. *"Fig. 4 no concerns, but descriptive."*

(no question or critique)

6. *"Fig. 5 shows very few changes in gene transcription between the control vs acute and acute vs adapted phases. There are other genes with greater L2FC than HORMAD1. Which ones are these? Do HORMAD1 levels and the levels of the other genes with high L2FC decrease in the adapted cells? There are no experiments addressing whether the increase in HORMAD1 levels have any functional significance. Are they related to the three phases of cell proliferation described above? If so, how?"*

Supplementary Table 1 shows all the genes with significant changes in the RNA-seq experiment. HORMAD1 is the protein-coding gene that was induced the most

relative to control, and the few mRNAs that change more are pseudogenes or readthroughs of other genes. As stated above in General Statement and in response to Reviewer 1 #1, we have taken Reviewer 2's advice and de-emphasized the potential contribution from *HORMAD1* expression. These changes involved revisions to the text, and to Figures 5 & 6 and Supplemental Figure S6.

7. *"What is the take-home message of this study? It seems very preliminary."*

We have added additional language in the introduction and discussion to highlight the uniqueness of the study (pages 4 and 15). We agree that the study is largely descriptive, but we argue that this in-depth characterization of the entire adaptation process is unique and valuable, and we have added a new mechanism of origin licensing inhibition to our understanding of the relationship between CDK activity and origin licensing inhibition.

****Referees cross-commenting****

*"The other reviewers also state that the study is descriptive and that the link to *HORMAD1* is merely a correlation."*

Reviewer 2 - Second review

"1. This manuscript has two main conclusions:

- a. Cyclin E overproduction induces replication stress, because it leads to underlicensing of DNA replication origins.*
- b. Adaptation to cyclin E overproduction is not just a reversal of the overproduction itself, because, according to the authors, adapted cells still overproduce cyclin E.*

In regard to the first conclusion, the authors monitored chromatin-bound MCM, whose levels are reported to decrease in cells overexpressing cyclin E. I previously commented that the decrease in bound MCM was variable from experiment to experiment. The authors chose new data to show less variability, but there is still variability. The data in Fig 2D-left panel show that only half the cells have reduced levels of bound MCM; in Figs 2E and 3E

(previously 3F?) almost all the cells have decreased levels of bound MCM. I will accept that there are reduced bound levels of MCM. However, whether this decrease affects origin firing remains to be determined, since it is thought that much more MCM is loaded than is needed for normal levels of origin firing. If origin firing is not affected, then the decrease in bound MCM is unlikely to explain the presence of replication stress.

*In regard to the second conclusion, it is clear that cyclin E levels decrease significantly over time (see Fig 3B, left: 2 days vs 30+ days and also Fig 4B: *cycE* 3 and 33 days). The authors claim that this decrease in cyclin E levels is not important, but they provide no justification for this claim. The most simple explanation of the results is that cyclin E levels decrease over time, which is a general problem with stable clones for inducible protein expression, and that this decrease explains the adaptation. Therefore, I am not convinced that the second conclusion is supported by the data."*

1a. We did not conduct origin firing assays per se, but the relationship between reducing origin licensing and ensuring replication stress is well-documented in the literature using both cell culture and mouse models. In addition, acute cyclin E induction causes a strong increase in S phase length of more than 4 hours (Figures 1C and 3D). Longer S phase is consistent with fewer origins firing or slower fork speed. In our previous work, we also showed that the rate of DNA synthesis is slowed by cyclin E overproduction in this same cell line (Matson 2017). The known direct role of cyclin E/CDK2 in origin firing but no reported cyclin E function in replication fork speed control, argues for reduced origin firing as the more likely explanation for slow replication and longer S phase... We have now made note of this point specifically on page 6

1b. Indeed, although cyclin E mRNA and protein levels decrease, cells are still substantially overproducing cyclin E relative to endogenous levels 30 days after initial induction with continuous doxycycline treatment. Importantly, the cyclin E-associated protein **kinase activity** is just as high as it had been during the initial induction (Figure 3C.) The fact that kinase activity was not reduced in adapted cells was also noted by Reviewer 1. We interpret the same kinase activity as an indicator that the other contributors to cyclin E/CDK2 enzymatic activity (CDK2, CAK etc.) were beyond saturated during the initial days of induction. Thus less *CCNE1* mRNA and less protein was still enough to maintain the same amount of kinase activity in adapted cells.

Despite this high CDK activity though, cells were proliferating very well, with near-normal G1 length, instead of descending into proliferative crisis from a truncated G1 as they did after initial induction; as stated in the text, there was no further change in proliferation after 30 days. These cells have persistently altered gene expression profiles that are not confined to *CCNE1* mRNA (Figure 5B). Adapted cells also have an intermediate licensing phenotype that takes weeks of growth without doxycycline to revert to normal (Figure 3E), despite cyclin E levels returning to normal within 48 hours of doxycycline withdrawal (Figure 3A, inset). We have added additional text to make these points even clearer on page 7.

2. *“When replying to the comments of the reviewers it is best to repeat the entire comment of the reviewer without any omissions or changes. Otherwise, it is impossible for one reviewer to understand the comments of the other reviewers. The authors summarized in their own words the comments of the reviewers.”*

We have now directly quoted each reviewer’s full comments in this response. We assumed from the cross-commenting that all reviewers had easy access to one another’s comments. We’ve typically had access to full comments in our own work as reviewers, and our motivation was simply to be concise.

Reviewer 2 Comments for the Author:

“The main difficulty with a potential revision is to provide data that prove or disprove the second conclusion.”

see response to point 1b above.

Reviewer 3 - First review (Evidence, reproducibility and clarity (Required): *“It has been known for a long time that overexpression of cyclin E shortens G1 and induces cells to enter S phase with a reduced number of licensed replication origins, resulting in replicative stress. This effect is of interest because cyclin E is overexpressed in some cancer cells. In the current paper, Limas et al investigate overexpression of cyclin E and make several observations that provide a clearer understanding of how it affects the cell cycle. They show that whilst cyclin E overexpression in immortalised RPE1 cells initially results in reduced cell proliferation, over a period of weeks the cells adapt and proliferation rates recover. Single-cell analysis shows that the acute reduction in proliferation is, as expected, associated with a shortened G1 and a lengthened S phase; surprisingly the adapted cells recover a more normal cell cycle pattern. The adapted cells have not suppressed cyclin production with associated kinase activity being still as high as in the acutely affected cells. The authors argue that this adaptation is not due to the expansion of rare mutant clones but is a non-genetic response of a large proportion of the population as it happens over a relatively short period to relatively small cell cultures. The molecular explanation for this adaptation remains unclear, but transcriptomic analysis shows that it is associated with expression of the *HORMAD1* gene. The authors also show a good correlation between cyclin E and *HORMAD1* expression in different cancer cells. The authors also provide arguments that the effect of underlicensing is an indirect effect of cyclin E mediated by it pushing cells prematurely into S phase, where other activities, such as cyclin A-CDK2, lead to suppression of the licensing process by known mechanisms.*”

The experiments are generally of a high quality and provide good support for the arguments being made. I have a few minor criticisms and suggestions for improvement, but nothing major in this regard (see minor points below).”

****Minor points****

1. *“1st paragraph of the Introduction: 'In G1 phase, cyclin D activates CDK4 and CDK6 to stimulate E2F-dependent transcription of a cohort of genes necessary for S phase entry.' I know this is just a summary to get readers up to speed, but the description should be consistent with the known role of cyclin E in activating E2F. One possible re-writing would be to say: 'In G1 phase, cyclin D activates CDK4 and CDK6 to initiate the stimulation of E2F-dependent...’”*

We have modified and expanded this information in the introduction, page 3.

2. *“The Mendez and Stillman 2000 citation is a bit of a strange choice alongside Evrin 2009 (note spelling throughout, not Ervin) and Remus and Diffley, 2009. Gillespie et al 2001 would be a better citation for the 4 licensing activities.”*

We have changed this reference on page 3.

3. *“1st paragraph of Results: 'Immunoblotting for each G1-S cyclin showed no effect of overproducing one cyclin on expression of another cyclin'. That doesn't appear to be true: In Fig 1 A overexpression of cyclin A causes increased expression of cyclin E -*

that's exactly what I would expect if cyclin A causes maximal Rb phosphorylation and hence maximal activation of E2F. The text should be amended to describe the figure more accurately."

We agree that a change is apparent in this example figure, though it was not the consistent result. We have modified the text to note this, page 5.

4. *"Top of page 5: I can't see the data in the supplemental figure 1A-B about doubling time as is stated in the main text. To use doubling time to estimate phase length depends on there being no change to the rate of cell death - was that checked?"*

We agree that if there had been substantial cell death it would impact our calculations of doubling times. However, we observed no evidence of cell death during adaptation (no increase in floaters, for example). We have made that point explicit in the results section about calculating cell cycle phase lengths using population doubling times, page 5.

5. *"Overproducing cyclin E only increases its kinase activity ~2 fold (Fig 1E), but expression levels of cyclin E are increased by ~10 fold (Fig 1B). I think that difference would be worth pointing out in the main text, as it is consistent with compensatory regulation of kinase activity for example by p21."*

We have modified the text to call attention to this difference and included additional mention of the potential for feedback restriction of CDK activity by the induced p21 in the results description for Figure 4 (p21 induction), pages 8-9.

6. *"Fig 2C is not quite as clear as it could be (for example, Supplemental Figure 1C looks better). Either replace it with cleaner data or perhaps cut the y axis at 100."*

we have made this replacement.

7. *"Figure 2E - how was this data normalised? I'd expect the exact MCM signal to vary between experimental replicates, and I presume the control values were normalised to 100 - if so, the figure legend should state this)"*

We did not normalize these data and reported the raw signal intensities in arbitrary units. It is a coincidence that the control intensities are centered ~100 arbitrary units which may have implied normalization; we have emphasized the raw intensity plots in the text page 5 and figure legend.

8. *"Figure 2E - I think it would be worth bringing the charts showing mean MCM loading for cyc D and cycA from Supplemental Figure 1D and 1F into main Figure 2. The difference in effect between the three cyclins is less extreme than I might have imagined. I think the text overstates the effect of cyclin E: 'By this analysis, cyclin E overproduction reduced licensing in early S by as much as ten-fold relative to control cells.' It would be fairer to say 'By this analysis, cyclin E overproduction reduced licensing in early S by approximately 3-fold.'"*

We did show a particularly dramatic example in original Figure 2C and 2D which was nearly 10-fold different, and our text did say “as much as ten-fold.” We acknowledge that the average effect is less as indicated in Figure 2E which is the combined data from 20 independent experiments, and we have modified the text on page 5.

We also took Reviewer 3’s suggestion to bring some data about licensing in cells overproducing cyclin D or cyclin A into Figure 2D for comparison.

9. *“Figure 2F: the number of 53BP1 nuclear bodies would be a more direct way to demonstrate the effect of under-licensing than gamma-H2AX foci.”*

We agree that 53BP1 nuclear bodies are a good marker, and indeed, we used that in Figure 4A. We elected to use γ -H2AX because we can quantify signals in many thousands of cells by flow cytometry instead of a few hundred by immunostaining for 53BP1. We now foreshadow that 53BP1 immunostaining is in Figure 4A in the paragraph describing Figure 2F, page 6.

10. *“Figure 3A and Supplemental Figure 3A: the unit on the y axis data is not adequality described in the legend - I'm guessing this is 1/hours, even though the x axis is in days. It might be more intuitive to use in the same units in both (hours or days). Was there any change in cell death during the experiment, as that would influence calculations of the doubling time?”*

We have both corrected the legend and defined the y-axis more clearly (page 7, and Figure 3A legend).

11. *“Figure 3A, inset. The legend doesn't really describe this experiment properly; the main text indicates that the '-' samples are from cells initially treated with dox and then had the dox withdrawn, but for how long is not clear.”*

We have corrected this omission in the text, page 7, and Figure 3A legend.

12. *“Why is the 30+ day cyclin E-induced decrease in MCM2 loading so much more pronounced in Figure 3F (looks ~80% reduction after 34 days) than 3E (looks ~20% reduction after 30+ days)?”*

The degree of underlicensing can vary within biological replicates performed at different times and in different weeks, but the underlicensing itself is consistent. We have replaced Figure 2C and 2D with a less dramatic example and revised the text to make it more clear that there are ranges of underlicensing. We also replaced the original Figures 3E and 3F with histograms from a different replicate to make it easier to compare different timepoints.

13. *“Figure 4A - the large 53BP1 foci are called 'Nuclear Bodies' (Lukas et al, 2011)”*

We have made this change throughout the relevant results section beginning on page 8.

(Significance (Required)):

"Whilst the paper clearly advances our understanding of the effect of cyclin E overexpression, the major weakness of the work is that no really new mechanistic understanding is given. The adaptation is an interesting effect and could well represent events occurring in the development of cancers associated with cyclin E overexpression, but it remains rather mysterious. The associated expression of HORMAD1 is intriguing and worthy of further investigation, but at present it remains simply a correlation. HORMAD1 has been linked to increased tolerance of DNA breaks so it would have been to know, for example, if knocking down HORMAD1 reduces the adaptation of cells to cyclin E overexpression."

****Referees cross-commenting****

"To be clear, I think the HORMAD1 data, even though correlative, is interesting and is appropriately described by the text. If the authors want to increase the impact of the work, they could do experiments that all 3 reviewers have suggested to show its functional importance; however, if the authors don't want to spend the time doing this and are content with having less impact, I think the manuscript is publishable in approximately its current form, so long as the other relatively minor issues are addressed."

Reviewer 3 - Second review

"JC Limas et al investigate the adaptation of human epithelial cells (RPE1-hTERT) to the overexpression of cyclin E1, a frequent occurrence in cancer cells. Following a short burst of proliferation, RPE1-hTERT cells with high levels of CycE1 undergo a proliferative crisis until some cells eventually adapt and manage to proliferate slightly faster than control cells without losing the overexpression of CycE1. The manuscript is focused in the impact of cycE1 on the licensing of replication origins in the G1 phase of the cell cycle. The proliferative crisis is characterized by a shortened G1 and premature entry into S phase with incomplete origin licensing. Profiling of gene expression by RNA-seq is performed at several time points across the process, finding relatively few genes or pathways that are differentially expressed across the experimental timeline. CycE1 overexpression correlates with the downregulation of a cohort of G1-expressed genes, but no causality is proven."

This is a very interesting research topic. My main criticism is that, while the study nicely recapitulates and confirms previous findings, it fails to provide new mechanistic insights."

Reviewer 3 Comments for the Author:

"In an earlier version of this manuscript, the consensus of four reviewers at Review Commons was that the study was "more descriptive than mechanistic", and a specific recommendation from all of us was to focus on the role of HORMAD1, a promising target derived from the RNA-seq experiments that was prominently featured in the Discussion. It seemed relatively straightforward to test at least how HORMAD1 downregulation affected the adaptation of cells to CycE1 overexpression."

Somehow surprisingly, the authors have chosen to "de-emphasize HORMAD1" in the revised manuscript (one of the four reviewers had mentioned that in the absence of more data, the emphasis on HORMAD1 should be downplayed). The "de-emphasizing" has been quite drastic

as this gene is not mentioned even once in the revised manuscript. Besides the *HORMAD1* obliteration, the current Figures are almost identical to those of the previous version.

As mentioned in response to Reviewer 1's original point 1, the fold change for *HORMAD1* was the highest of any protein-coding gene other than those encoding cyclin E and p21... but the absolute level of mRNA was actually quite low – much less than tumor cell lines that overproduce *HORMAD1* in fact. Despite numerous attempts with multiple antibodies, we were unable to detect *HORMAD1* protein in RPE cells which hampers the downregulation experiments.

The issue remains that the main phenotypes linked to cycE overexpression in this system (shorter G1 phase, premature entry into S phase, and origin under-licensing) recapitulate previous findings that can be tracked back to Ekholm-Reed et al, J Cell Biol 2004, and other subsequent studies including at least one from the Cook lab (Matson et al, eLife 2017). The authors argue (Letter of response to reviewers) that Ekholm-Reed et al (2004) "did not show effects on proliferation, genome stability, or consequences of long-term cyclin E expression in the transformed cell lines because the adenoviral gene delivery method only induces transient overproduction"; but this is because the 2004 JCB study was itself a mechanistic follow-up of a previous article (Spruck et al, Nature 2009 [author note– 1999, not 2009]) that presented evidence of chromosomal instability after long-term (30 days) overexpression of CycE.

The studies mentioned are relevant (and cited), but there are still differences from our current work. Ekholm-Reed and Matson both examined licensing but not during prolonged cyclin E overproduction, just 1-2 cell cycles. Spruck characterized long-term cyclin E expression but did not analyze origin licensing - not at all surprising given the state of the field in 1999. We are the first to analyze licensing during and after long-term cyclin E overproduction, and we've had added additional clarification about these differences in the Discussion on page 13.

In summary, this is a solid but mainly descriptive study from a well-known and respected group in the DNA replication field. As reviewers for J Cell Sci are specifically asked to assess whether research articles "make a significant and novel contribution to our understanding of cell biology, (are) of broad interest to the cell biology community, and provide mechanistic insight", my opinion is that the manuscript falls a bit short on conceptual novelty and mechanistic advance. Its suitability for J Cell Sci is for the Editors to decide."

We acknowledge that degrees of significance, novelty, and defining what constitutes mechanistic insight are subjective determinations.

Reviewer 4 – First review

(Evidence, reproducibility and clarity (Required)): *"In this study from the JC Cook lab, Limas et al investigate how non-cancerous, immortalized human epithelial cells (RPE1-hTERT) may adapt to the chronic overexpression of cyclin E1. They analyze how cells behave in terms of proliferation rate and origin licensing capacity, i.e. the recruitment of MCM proteins to DNA in order to initiate DNA replication. Following a short burst of rapid proliferation, RPE1-hTERT cells expressing high levels of CycE undergo a proliferative crisis characterized by incomplete origin*

licensing and very slow cell cycles. Some cells eventually adapt to the high CycE levels and end up proliferating slightly faster than control cells, while origin licensing is partially recovered. The authors perform RNA sequencing at different time points across the process, trying to identify a signature of transcriptional changes that could explain the adaptation. A promising target is found (HORMAD1), and computational analyses reveal that the levels of HORMAD1 also correlate with those of CycE1 in basal-like breast cancers.”

****Major points:****

1. *“The study is well designed and the experiments seem quite logical as a first attempt to understand the -presumably complex- mechanisms underlying the adaptation of normal cells to grow in the presence of high levels of CycE. The “adaptation phenotypes” are well described in Figures 1-4 (even if some observations are confirmatory of previous studies, some of them published in the 1990s and 2000s). However, in my opinion the study somehow falls short in terms of identifying the possible adaptation mechanism(s). The transcriptomics approach shown in Figure 5B reveals global changes in G1-expressed genes that could be both cause or consequence of the adaptation, as properly acknowledged by the authors in the text. The analyses shown in Figure 5C and 5D reveal few changes that pass the statistical filters and make clear sense as a whole. This leaves HORMAD1 as the most promising candidate, but its role is not validated (see below)”*

We are also curious about the molecular mechanisms driving these phenotypic changes. As noted in the General Statements section, we assert that deep characterization of the adaptation process itself is novel, and that valuable insights in the field have often come from what are essentially descriptive studies. Given that the mechanisms of adaptation involve multiple changes, we predict that testing the functions of those mechanisms will be another full study. We agree that this finding is preliminary and have de-emphasized HORMAD1 in this manuscript.

2. *“While I agree that some of the authors' observations are not consistent with genetic mutations driving adaptation, several parameters are being monitored simultaneously (duration of G1, origin licensing, rate of cell division) and it is conceivable that genetic mutations may drive or at least influence some of them, particularly the ability to overcome the proliferation crisis. In this regard, the adaptation to high CycE levels has parallels with the well-studied cell immortalization process.”*

We agree, and we acknowledged this possibility in the discussion. We have now added more text in the results to this effect also (page 8 and page 15). Genetic changes from oncogene expression are expected, but our evidence for non-genetic changes is strong and based on the arguments in the Discussion, page 15. We observed proliferation at all time points throughout the dish instead of rare colonies, the adaptation occurred within only ~2-3 weeks regardless of how many cells we started with, the adapted cells showed moderate underlicensing that reverted only slowly over several weeks upon dox withdrawal (Figure 3E) even though cyclin E levels returned to normal within 2 days (Figure 3A).

3. *"The identification of **HORMAD1** gene upregulation is interesting because this gene is preferentially meiotic, and the authors find other instances in which **HORMAD1** expression correlates with the levels of **CCNE1** (e.g. breast cancer). Indeed, a large part of the discussion is focused on **HORMAD1**. However, the causal role of **HORMAD1** remains speculative at this point and it would require some fundamental molecular characterization, e.g. could adaptation occur in its absence?, etc."*

We have reduced the speculation about **HORMAD1** in favor of the combination of changes that promote G1 lengthening to support more effective origin licensing.

****Minor points:****

1. *"Figure 1A. The text argues that ectopic expression of one Cyclin does not influence the levels of the others. While this would not be a critical issue, I believe some fluctuations may be obscured by the long-exposure blots shown (particularly **cycE** in the left column (after cyclin D o/e), and **cycD** in the right column (after cyclin A o/e)."*

We quantified band intensities using a variety of exposures, and we chose these darker ones because they show the endogenous levels of cyclin E for comparison. Our imaging system indicates when signals are saturated, and we did not include any images with signals outside the linear range of detection; we have added this information on pages 4-5, and in the methods subsection on immunoblotting. We have also added the lighter exposures as Supplemental Figure S6.

2. *"Figure 2F. The conclusion that cells overexpressing **cycE** are hypersensitive to gemcitabine seems derived from the statistical significance of the difference between lanes 2 and 4. However, the fold-increase in **γH2AX** caused by GEM in control cells (lanes 1-3) is similar, or probably higher, than in **cycE**-overexpressing cells. In order to demonstrate defects in dormant origin activation, the use of stretched DNA fibers would be a more direct approach."*

We have revised Figure 2F to show the gemcitabine treatment primarily as a positive control for **γ-H2AX** signal and cyclin E overproduction for comparison. The conclusion we draw is that cyclin E overproduction induces endogenous replication stress, and we cite studies where direct origin licensing inhibition also induces **γ-H2AX** in support of this assay. The suggestion to conduct single molecule DNA fiber analysis to compare dormant origin usage is beyond the focus of this paper which is the adaptation process rather than more characterization of the replication stress phenotype; it's well-established that reduced origin licensing leads to fewer dormant origins. We are careful to communicate that reduced dormant origin availability is a potential source of the stress without claiming to have directly measured them.

3. *"In the Introduction (p.4, l.4) the authors state that "some tumor-derived cell lines with high **CycE** proliferate without experiencing similar levels of replication stress or impaired origin licensing (Ekholm-Reed et al, 2004,...). However, the study of Ekholm-Reed et al (2004) shows precisely that even carcinoma cells display impaired origin licensing upon **cycE** overexpression."*

We cited this paper because it has a nice example immunoblot of endogenous cyclin E expression in several cancer cell lines directly compared to a non-transformed control. However, as the title of the paper states, even these cancer cells showed detectably lower licensing levels when ectopic cyclin E was briefly expressed on top of the naturally high endogenous levels. That earlier study did not show effects on proliferation, genome stability, or consequences of long-term cyclin E expression in the transformed cell lines because the adenoviral gene delivery method only induces transient overproduction. We have used alternative citations to support the conclusion that some cancer cell lines proliferate with high levels of endogenous cyclin E in the ranges we achieve in this study on page 4. We cite Ekholm-Reed in the discussion regarding the current paradigm of direct CDK-mediated licensing inhibition.

Reviewer #4

*“The study is pertinent as part of a long-time effort by several laboratories to understand the effects of abnormal *cycE* levels in genomic stability, which was started by the pioneering work of Steve Reed and others in the mid 1990s. The topic is very interesting and should be appealing to scientists interested in cell cycle, CDKs, DNA replication and cell transformation by oncogenes.*

*As mentioned before, some of the observations about the cellular response to *cycE* overexpression recapitulate previous findings (e.g. shorter G1 phase, origin under-licensing, replicative stress), even if some technical approaches are new and the cell lines are different. The most original part is the transcriptional profiling of the adaptation process, which still sounds preliminary. I believe the manuscript would greatly improve if the participation of *HORMAD1* could be demonstrated directly.”*

Reviewer 4 - Second review

(Only three reviewers responded to the first revision. We assume they were the original Reviewers 1-3.)

January 24, 2022

RE: Life Science Alliance Manuscript #LSA-2022-01378

Dr. Jeanette Gowen Cook
University of North Carolina at Chapel Hill
Biochemistry and Biophysics
Campus Box 7260, 3010 Genetic Medicine Building
120 Mason Farm Rd.
Chapel Hill, NC 27599-7260

Dear Dr. Cook,

Thank you for submitting your revised manuscript entitled "Quantitative Profiling of Adaptation to Cyclin E Overproduction". We would be happy to publish your paper in Life Science Alliance pending final revisions necessary to meet our formatting guidelines.

- please upload your main manuscript text as an editable doc file
- Please upload all figure files as individual ones, including the supplementary figure files; all figure legends should only appear in the main manuscript file
- please add the Twitter handle of your host institute/organization as well as your own or/and one of the authors in our system
- please add callouts for Figures 5D and S6A-C to your main manuscript text

FIGURE CHECKS:

- please include sizes next to all blots
- please include a scale bar in Figure S4A, and indicate its size in the Legend
- In Figure S6A, the cyclin A blot at 20ng/ML dox appears to have a box around it. Was something placed over the blot, or is this just an artifact?

A. FINAL FILES:

B. MANUSCRIPT ORGANIZATION AND FORMATTING:

Sincerely,

January 31, 2022

RE: Life Science Alliance Manuscript #LSA-2022-01378R

Dr. Jeanette Gowen Cook
University of North Carolina at Chapel Hill
Biochemistry and Biophysics
Campus Box 7260, 3010 Genetic Medicine Building
120 Mason Farm Rd.
Chapel Hill, NC 27599-7260

Dear Dr. Cook,

Thank you for submitting your Research Article entitled "Quantitative Profiling of Adaptation to Cyclin E Overproduction". It is a pleasure to let you know that your manuscript is now accepted for publication in Life Science Alliance. Congratulations on this interesting work.

*****IMPORTANT:** If you will be unreachable at any time, please provide us with the email address of an alternate author. Failure to respond to routine queries may lead to unavoidable delays in publication. *******

DISTRIBUTION OF MATERIALS:

Again, congratulations on a very nice paper. I hope you found the review process to be constructive and are pleased with how the manuscript was handled editorially. We look forward to future exciting submissions from your lab.

Sincerely,
